# Saliency-Aware Neural Architecture Search

**Ramtin Hosseini and Pengtao Xie**
UC San Diego
`rhossein@eng.ucsd.edu, p1xie@eng.ucsd.edu`

## Abstract

Recently a wide variety of NAS methods have been proposed and achieved considerable success in automatically identifying highly-performing architectures of neural networks for the sake of reducing the reliance on human experts. Existing NAS methods ignore the fact that different input data elements (e.g., image pixels) have different importance (or saliency) in determining the prediction outcome. They treat all data elements as being equally important and therefore lead to suboptimal performance. To address this problem, we propose an end-to-end framework which dynamically detects saliency of input data, reweights data using saliency maps, and searches architectures on saliency-reweighted data. Our framework is based on four-level optimization, which performs four learning stages in a unified way. At the first stage, a model is trained with its architecture tentatively fixed. At the second stage, saliency maps are generated using the trained model. At the third stage, the model is retrained on saliency-reweighted data. At the fourth stage, the model is evaluated on a validation set and the architecture is updated by minimizing the validation loss. Experiments on several datasets demonstrate the effectiveness of our framework.

## 1 Introduction

Neural architecture search (NAS) [76, 41, 47], which aims to automatically identify highly performant neural architectures, has received much attention recently. Existing NAS methods treat all elements in an input data example (such as pixels in an image, tokens in a sentence, etc.) as being equally important, without considering the different saliency of individual elements, which leads to less-optimal performance. In machine learning applications, an input data example typically consists of many data elements. For instance, an image example consists of a grid of pixel elements and a sentence example consists of a sequence of token elements. When used to make a prediction, different data elements have different importance (or saliency). For example, when predicting which object category an image belongs to, pixels in foreground object regions are more important than those in background regions. Such saliency information is not leveraged by existing NAS methods.

In this paper, we aim to bridge this gap, by proposing a saliency-aware NAS method which automatically identifies the saliency of data elements and leverages that to search for better architectures. Our framework is formulated as a four-level optimization problem. At the first level, the model tentatively fixes its architecture and trains its first set of network weights. At the second level, the trained model generates saliency maps using an adversarial attack based method [18]. At the third level, input data is reweighted using saliency maps and the second set of model weights are trained using reweighted data. At the fourth level, the two sets of trained model weights are evaluated on a human-provided validation set and the architecture is optimized by minimizing the validation losses.

The major contributions of this paper are as follows:

- We propose a framework which performs end-to-end detection of saliency and leverage saliency-reweighted data to improve neural architecture search.

36th Conference on Neural Information Processing Systems (NeurIPS 2022).

- We formulate our method as a multi-level optimization problem which performs model training on unweighted data, saliency map generation, model retraining on saliency-reweighted data, and architecture update in a unified manner.

## 2 Related works

**Neural architecture search (NAS).** Early NAS approaches [76, 45, 77] are mostly based on reinforcement learning (RL) which use a policy network to generate architectures and evaluate these architectures on validation sets. The validation loss is used as a reward to optimize the policy network and train it to produce high-quality architectures. Differentiable search methods [5, 41, 65] parameterize architectures as differentiable functions and perform search using efficient gradient-based methods. In these methods, the search space of architectures is composed of a large set of building blocks where the output of each block is multiplied with a smooth variable indicating how important this block is. Under such a formulation, search becomes solving a mathematical optimization problem defined on the importance variables where the objective is to find an optimal set of variables that yield the lowest validation loss. This optimization problem can be solved efficiently using gradient-based methods. Differentiable NAS research is initiated by DARTS [41] and further improved by subsequent works such as P-DARTS [11], PC-DARTS [66], etc. P-DARTS [11] grows the depth of architectures progressively in the search process. PC-DARTS [66] samples sub-architectures from a super network to reduce redundancy during search. Our proposed framework is orthogonal to existing NAS methods and can be used in combination with any differentiable NAS method to further improve these methods. Besides RL-based approaches and differentiable NAS, another paradigm of NAS methods [40, 47] are based on the evolutionary algorithm where architectures are formulated as individuals in a population. High-quality architectures produce offspring to replace low-quality architectures, where the quality is measured using fitness scores.

**Saliency detection.** Many methods [70, 75, 56, 44, 51, 50, 46] have been proposed for saliency detection, based on perturbing inputs [70, 75], propagating gradients [3, 55, 56], attention [37, 69, 44], model approximation [49, 2], etc. Several works [51, 50, 46] show that leveraging saliency of input data can enhance model's predictive power. Rieger et al. [50] leverage domain-specific rules or knowledge to provide "groundtruth" saliency. Such rules/knowledge are difficult to obtain in many applications. Pillai and Pirsiavash [46] encourage a prediction model to produce saliency maps that are consistent with those generated by GradCAM [53]. GradCAM is an unsupervised approach; without any supervision from humans, its generated saliency maps may not be reliable. For example, it is shown in [57] that GradCAM cannot highlight adversarial image patches that cause wrong predictions. In [51], saliency maps are either labeled by humans or auto-generated based on gradient magnitude with no human supervision, which suffers the same problems as [53, 50]. Different from existing works, our method generates saliency maps with weak supervision such as human-provided class labels. Such weak supervision is much easier to obtain than human annotations of saliency maps and can yield more reliable saliency maps than using no supervision at all.

**Bi-level optimization.** Many ML methods [19, 4, 20, 41, 54, 72] have been formulated as bi-level optimization (BLO) problems. In these methods, network weights are learned by solving an inner optimization problem defined on a training set while meta parameters are learned by solving an outer optimization problem defined on a validation set. BLO-based methods have been applied for neural architecture search [41], hyperparameter tuning [19], learning rate adaptation [4], data selection [54, 48, 64], meta learning [20], label correction [72], etc., where meta parameters are neural architectures, hyperparameters, importance weights of data examples, meta network weights, etc. Many optimization algorithms [14, 21, 23, 32, 42, 67] have been developed for solving BLO problems where convergence analysis is provided.

## 3 Methods

In this section, we propose a four-level optimization based framework to perform saliency-aware neural architecture search. For the ease of presentation, we assume the task is image classification. In the experiments, we show that our method can be applied for other tasks as well.

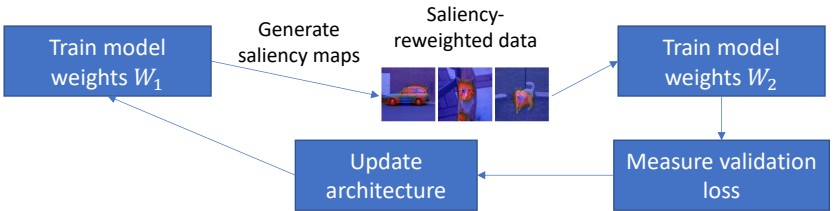

Figure 1: Overview of our framework.

## 3.1 A four-level optimization framework

In our framework (Figure 1), a model has a learnable architecture $A$ and two sets of learnable network weights $W_1$ and $W_2$. $W_2$ is a tensor that has the same dimensions as $W_1$. The weight values in $W_2$ and $W_1$ are different. It consists of four stages performed end-to-end. At the first stage, the model trains its network weights $W_1$ with the architecture $A$ tentatively fixed. At the second stage, the trained $W_1$ generates saliency maps for input images: a saliency score is calculated for each pixel. At the third stage, images are reweighted using saliency maps and saliency-reweighted images are used to train model weights $W_2$. At the fourth stage, the trained $W_2$ is evaluated on a human-labeled validation set and the architecture $A$ is updated by minimizing the validation loss.

**Stage I.** At the first stage, the model trains its first set of network weights $W_1$ by minimizing a loss $L$ on training dataset $D^{(\mathrm{tr})}$, with the architecture $A$ tentatively fixed:

$$W_1^*(A) = \mathrm{argmin}_{W_1} \, L(W_1, A, D^{(\mathrm{tr})}). \tag{1}$$

To define the training loss, we need to use both the architecture parameters $A$ and network weights $W_1$. However, $A$ cannot be updated by minimizing the training loss. Otherwise, a trivial solution of $A$ will be yielded: $A$ can perfectly overfit the training data but will make incorrect predictions on unseen data examples. $W_1^*(A)$ denotes that the optimal weights $W_1^*$ depends on $A$. This is because $L(W_1, A, D^{(\mathrm{tr})})$ is a function of $A$, and $W_1^*$ depends on $L(W_1, A, D^{(\mathrm{tr})})$.

**Stage II.** At the second stage, the trained $W_1^*(A)$ is used to generate saliency maps. Specifically, given an input image $x$, we first use $W_1^*(A)$ and $A$ to predict the class label (denoted by $f(x; W_1^*(A), A)$) of $x$. Then an adversarial attack based approach [22, 18] is leveraged to calculate saliency scores of pixels. Adversarial attack adds small random perturbations $\delta$ to pixels in $x$ so that the prediction outcome on the perturbed image $x + \delta$ is no longer $f(x; W_1^*(A), A)$. Pixels perturbed more are more correlated with the prediction outcome $f(x; W_1^*(A), A)$ and are considered to be more salient. This process amounts to solving the following optimization problem:

$$\{\delta_i^*(W_1^*(A), A)\}_{i=1}^N = \mathrm{argmax}_{\{\|\delta_i\|_\infty \le \varepsilon\}_{i=1}^N} \, \sum_{i=1}^N \ell(f(x_i + \delta_i; W_1^*(A), A), f(x_i; W_1^*(A), A)) \tag{2}$$

where $\delta_i$ is the perturbation added to image $x_i$ and $\varepsilon$ is a small norm-bound. $N$ is the number of images. $f(x_i + \delta_i; W_1^*(A), A)$ and $f(x_i; W_1^*(A), A)$ are predictions made by $W_1^*(A)$ and $A$ on $x_i + \delta_i$ and $x_i$. Assume the number of classes is $K$. $f(x_i + \delta_i; W_1^*(A), A)$ and $f(x_i; W_1^*(A), A)$ are $K$-dimensional vectors containing prediction probabilities on individual classes. $\ell(\cdot, \cdot)$ is the cross-entropy loss with $\ell(\mathbf{a}, \mathbf{b}) = -\sum_{k=1}^K b_i \log a_i$. In this optimization problem, we aim to find perturbations for each image so that the predicted outcome on the perturbed image is largely different from that on the original image. The learned optimal perturbations are considered as saliency scores of pixels: larger perturbations indicate that the corresponding pixels are more correlated with the prediction outcome and therefore are more salient. $\delta_i^*$ depends on $W_1^*(A)$ and $A$ since $\delta_i^*$ depends on the loss in Eq.(2), and the loss is a function of $W_1^*(A)$ and $A$.

**Stage III.** At the third stage, given the saliency scores $\{\delta_i^*(W_1^*(A), A)\}_{i=1}^N$, the second set of model weights $W_2$ are trained. We use the saliency scores to reweight the pixels: $x \odot \delta$, where $\odot$ denotes element-wise multiplication (we compare with other reweighting mechanisms in Table 4). Pixels that are more salient are given more weights. Then $W_2$ is trained on these weighted pixels:

$$W_2^*(\{\delta_i^*(W_1^*(A), A)\}_{i=1}^N, A) = \mathrm{argmin}_{W_2} \, \sum_{i=1}^N \ell(f(\delta_i^*(W_1^*(A), A) \odot x_i; W_2, A), t_i), \tag{3}$$

where $f(\delta_i^*(W_1^*(A),A) \odot x_i; W_2, A)$ is the prediction made by $W_2$ and $A$ on the weighted image $\delta_i^*(W_1^*(A),A) \odot x_i$, and $t_i$ is the class label. $W^*$ depends on $\{\delta_i^*(W_1^*(A),A\}_{i=1}^N$ and $A$ since $W^*$ depends on the loss in Eq.(3), and the loss is a function of $\{\delta_i^*(W_1^*(A),A)\}_{i=1}^N$ and $A$.

**Stage IV.** At the fourth stage, $W_2^*(\{\delta_i^*(W_1^*(A),A)\}_{i=1}^N, A)$ and $W_1^*(A)$ are evaluated on a human-labeled validation set $D^{(\text{val})}$. The architecture $A$ is updated by minimizing the validation losses:

$$\min_A L(W_2^*(\{\delta_i^*(W_1^*(A),A)\}_{i=1}^N, A), A, D^{(\text{val})}) + \gamma L(W_1^*(A), A, D^{(\text{val})}), \tag{4}$$

where $\gamma$ is a tradeoff parameter.

**Four-level optimization framework.** We integrate the four stages into a unified four-level optimization framework and obtain the following formulation:

$$\min_A L(W_2^*(\{\delta_i^*(W_1^*(A),A)\}_{i=1}^N, A), A, D^{(\text{val})}) + \gamma L(W_1^*(A), A, D^{(\text{val})})$$

$$s.t. \quad W_2^*(\{\delta_i^*(W_1^*(A),A)\}_{i=1}^N, A) = \text{argmin}_{W_2} \; \sum_{i=1}^N \ell(f(\delta_i^*(W_1^*(A),A) \odot x_i; W_2, A), t_i)$$

$$\{\delta_i^*(W_1^*(A),A)\}_{i=1}^N = \text{argmax}_{\{\|\delta_i\|_\infty \leq \varepsilon\}_{i=1}^N} \; \sum_{i=1}^N \ell(f(x_i + \delta_i; W_1^*(A),A), f(x_i; W_1^*(A),A))$$

$$W_1^*(A) = \text{argmin}_{W_1} L(W_1, A, D^{(\text{tr})}). \tag{5}$$

In this framework, there are four optimization problems, each corresponding to a learning stage. From bottom to up, the optimization problems correspond to learning stage I to IV respectively. The first three optimization problems are nested on the constraint of the fourth optimization problem. These four stages are conducted end-to-end in this unified framework. The solution $W_1^*(A)$ obtained at the first stage is used to generate explanations at the second stage. The saliency maps $\{\delta_i^*(W_1^*(A),A)\}_{i=1}^N$ obtained at the second stage are used to train $W_2$ at the third stage. The solutions obtained at the first and third stage are used to calculate validation losses at the fourth stage. The architecture $A$ updated at the fourth stage changes the training loss at the first stage and consequently changes the solution $W_1^*(A)$, which subsequently changes $\{\delta_i^*(W_1^*(A),A)\}_{i=1}^N$ and $W_2^*(\{\delta_i^*(W_1^*(A),A)\}_{i=1}^N, A)$.

**Optimization algorithm.** To solve the problem in Eq.(5), we used a standard algorithm developed in [41], which is broadly used in many previous works and demonstrated to be effective in the literature. The convergence analysis of this algorithm has been given in many works [21, 23, 32, 42, 67]. The optimization algorithm is not the focus or contribution of our work. In this algorithm, we calculate the gradient of $L(W_1, A, D^{(\text{tr})})$ w.r.t $W_1$ and approximate $W_1^*(A)$ using a one-step gradient descent update of $W_1$. We plug the approximation $W_1'$ of $W_1^*(A)$ into $\ell(f(x_i + \delta_i; W_1^*(A),A), f(x_i; W_1^*(A),A))$ and obtain an approximated objective denoted by $O_{\delta_i}$. Then we approximate $\delta_i^*(W_1^*(A),A)$ using a one-step gradient ascent update of $\delta_i$ based on the gradient of $O_{\delta_i}$. Next, we plug the approximation $\delta_i'$ of $\delta_i^*(W_1^*(A),A)$ into $\ell(f(\delta_i^*(W_1^*(A),A) \odot x_i; W_2, A), t_i)$ and get another approximated objective denoted by $O_{W_2}$. Then we approximate $W_2^*(\{\delta_i^*(W_1^*(A),A)\}_{i=1}^N, A)$ using a one-step gradient descent update of $W_2$ based on the gradient of $O_{W_2}$. Finally, we plug the approximation $W_1'$ of $W_1^*(A)$ and the approximation $W_2'$ of $W_2^*(\{\delta_i^*(W_1^*(A),A)\}_{i=1}^N, A)$ into the validation loss and get the third approximate objective $O_A$. $A$ is updated by descending the gradient of $O_A$. These steps iterate until convergence.

# 4 Experiments

In this section, we present experimental results. Please refer to the supplements for detailed hyperparameter settings and additional results (e.g., on inference costs, ablation studies).

## 4.1 Experiments on image classification

Following [41] the architecture $A$ is parameterized in a differentiable way. $A$ contains a set of importance weights, each multiplied to the output of a candidate architecture block. Architecture search amounts to learning these weights using gradient methods. After learning, blocks with top weights compose an architecture. Each experiment consists of two phrases: 1) architecture search where an optimal cell is identified, and 2) architecture evaluation where multiple copies of the optimal cell are stacked into a larger network, which is retrained from scratch.

Table 1: Test errors on CIFAR-100 (C100) and CIFAR-10 (C10), number of model parameters (in millions), and search cost (GPU days on a Nvidia 1080Ti). SANAS-darts2nd represents that SANAS is applied to DARTS-2nd. Similar meanings hold for other notations in such a format. * means the results are taken from DARTS$^-$ [12].

| Method | Error-C100 | Error-C10 | Param. | Cost |
|---|---|---|---|---|
| *ResNet [26] | 22.10 | 6.43 | 1.7 | - |
| *DenseNet [30] | 17.18 | 3.46 | 25.6 | - |
| *PNAS [39] | 19.53 | 3.41±0.09 | 3.2 | 150 |
| *ENAS [45] | 19.43 | 2.89 | 4.6 | 0.5 |
| *AmoebaNet [47] | 18.93 | 2.55±0.05 | 3.1 | 3150 |
| *GDAS [17] | 18.38 | 2.93 | 3.4 | 0.2 |
| *R-DARTS [71] | 18.01±0.26 | 2.95±0.21 | - | 1.6 |
| *DARTS$^-$ [12] | 17.51±0.25 | 2.59±0.08 | 3.3 | 0.4 |
| AutoFormer [8] | 17.42±0.17 | 2.72±0.09 | 3.7 | 2.8 |
| Sampling [24] | 17.30±0.10 | 2.75±0.11 | 3.8 | 2.0 |
| *DropNAS [27] | 16.95±0.41 | 2.58±0.14 | 4.4 | 0.7 |
| *DrNAS [10] | - | 2.54±0.03 | 4.0 | 0.4 |
| *ISTA-NAS [68] | - | 2.54±0.05 | 3.3 | 0.1 |
| *MiLeNAS [25] | - | 2.51±0.11 | 3.9 | 0.3 |
| *GAEA [38] | - | 2.50±0.06 | - | 0.1 |
| *PDARTS-ADV [9] | - | 2.48±0.02 | 3.4 | 1.1 |
| *Darts2nd [41] | 20.58±0.44 | 2.76±0.09 | 3.1 | 4.0 |
| EC-darts2nd [51] | 20.05±0.31 | 2.83±0.12 | 3.3 | 5.7 |
| CDEP-darts2nd [50] | 19.53±0.46 | 2.75±0.05 | 3.2 | 5.3 |
| GMPGC-darts2nd [46] | 19.08±0.36 | 2.81±0.07 | 3.2 | 5.6 |
| Ours-darts2nd | **16.42**±0.09 | **2.54**±0.05 | 3.2 | 4.6 |
| †Pcdarts [66] | 17.96±0.15 | 2.57±0.07 | 3.9 | 0.1 |
| EC-pcdarts [51] | 17.83±0.28 | 2.63±0.11 | 4.1 | 0.9 |
| CDEP-pcdarts [50] | 17.88±0.13 | 2.75±0.08 | 4.0 | 1.1 |
| GMPGC-pcdarts [46] | 17.73±0.09 | 2.64±0.05 | 4.0 | 1.0 |
| Ours-pcdarts | **16.19**±0.04 | **2.49**±0.03 | 3.9 | 0.8 |
| †Prdarts [73] | 16.48±0.06 | 2.37±0.03 | 3.4 | 0.2 |
| EC-prdarts [51] | 17.32±0.14 | 2.58±0.08 | 3.5 | 1.1 |
| CDEP-prdarts [50] | 16.86±0.07 | 2.54±0.05 | 3.4 | 1.3 |
| GMPGC-prdarts [46] | 16.37±0.10 | 2.46±0.06 | 3.6 | 1.1 |
| Ours-prdarts | **16.01**±0.03 | **2.30**±0.04 | 3.6 | 0.8 |
| †Pdarts [11] | 17.52±0.06 | 2.54±0.04 | 3.6 | 0.3 |
| EC-pdarts [51] | 17.25±0.11 | 2.68±0.07 | 3.8 | 1.3 |
| CDEP-pdarts [50] | 17.49±0.08 | 2.63±0.10 | 3.6 | 0.9 |
| GMPGC-pdarts [46] | 17.33±0.10 | 2.59±0.07 | 3.7 | 1.1 |
| Ours-pdarts | **15.16**±0.09 | **2.45**±0.03 | 3.6 | 1.1 |

**Datasets** We used three datasets: CIFAR-10 [35], CIFAR-100 [36], and ImageNet [15]. Both CIFAR-10 and CIFAR-100 contain 60K images from 10 classes. For each of them, we split it into a train, validation, and test set with 25K, 25K, and 10K images respectively. ImageNet contains 1.2M training images and 50K test images from 1000 objective classes. Following [66], we randomly sample 10% of the 1.2M images to form a new training set and another 2.5% to form a validation set, then perform a search on them.

**Experimental settings** We experimented with the search spaces in DARTS [41], P-DARTS [11], and PC-DARTS [66]. The tradeoff parameter $\gamma$ is set to 2. The norm bound $\varepsilon$ of perturbations is set to 0.03. During architecture search, for CIFAR-10 and CIFAR-100, the architecture of the target is a stack of 8 cells. Each cell consists of 7 nodes. Initial channel number is 16. The search algorithm was based on SGD, with a batch size of 64, an initial learning rate of 0.025 with cosine scheduling, an epoch number of 50, a weight decay of 3e-4, and a momentum of 0.9. We update the architecture $A$ every 5 mini-batches (iterations), update model weights $W_2$ and perturbations $\delta$ every 3 mini-batches,

and update $W_1$ on every mini-batch. The rest of hyperparameters mostly follow those in DARTS, P-DARTS, and PC-DARTS. We compare with the following baselines: 1) explanation constraints (EC) [51]; 2) contextual decomposition explanation penalization (CDEP) [50]; 3) global max pooling + GradCAM (GMPGC) [46]. The mean and standard deviation of 10 random runs are reported.

**Results and analysis on CIFAR-100 and CIFAR-10**  Table 1 shows results on CIFAR-100 and CIFAR-10. Applying our framework to DARTS, P-DARTS, and PC-DARTS, test errors are greatly reduced, which shows that by end-to-end detecting and leveraging saliency of pixels, the quality of searched architectures can be improved. Another observation from Table 1 is: the performance gain of our method does not come at the cost of substantially increasing model size (number of parameters) or search cost.

Table 1 shows that our methods outperform EC, CDEP, and GMPGC. These methods use GradCAM, gradient magnitude, and pretrained semantic segmentation models to calculate saliency scores, which are not very reliable. In contrast, the calculation of saliency in our method is weakly supervised by the validation loss of the second model calculated on human-provided class labels, which have higher fidelity. As analyzed earlier, saliency with higher fidelity can result in higher-quality architectures.

**Results on ImageNet**  In Table 2, we compare different methods on ImageNet. In experiments based on PC-DARTS, architectures are searched on a subset of ImageNet. In other experiments, architectures are searched on CIFAR-10 and CIFAR-100. Ours-darts2nd-cifar10 denotes that our method is applied to DARTS-2nd and performs search on CIFAR10. Similar meanings hold for other notations in such a format. The observations made from these results are consistent with those made from Table 1. The architectures searched using our methods are consistently better than those searched by corresponding baselines. These results again show that by end-to-end detecting and leveraging saliency can improve architecture search.

**Sanity check of saliency maps**  We evaluate saliency maps generated by the adversarial saliency method using model parameter cascading randomization tests [1]. The model architecture is searched by SANAS on ImageNet. Figure 2(left) shows that saliency maps change considerably as more layers are randomized, on multiple ImageNet examples. Figure 2(right) shows the Spearman rank correlation (with absolute values) between original saliency maps and randomized saliency maps, on ImageNet. The rank correlation consistently decreases as more layers are randomized. These results demonstrate that saliency maps generated by the adversarial saliency method are sensitive to model parameters and pass the sanity check.

Table 2: Top-1 and top-5 test errors on ImageNet in the mobile setting. Results marked with * are obtained from DARTS$^-$ [12] and DrNAS [10]. The rest notations are the same as those in Table 1.

| Method | Top-1 | Top-5 |
|---|---|---|
| *Inception-v1 [60] | 30.2 | 10.1 |
| *MobileNet [28] | 29.4 | 10.5 |
| *ShuffleNet $2\times$ (v2) [43] | 25.1 | 7.6 |
| *NASNet-A [77] | 26.0 | 8.4 |
| *PNAS [39] | 25.8 | 8.1 |
| *MnasNet-92 [61] | 25.2 | 8.0 |
| *AmoebaNet-C [47] | 24.3 | 7.6 |
| *PARSEC-CIFAR10 [6] | 26.0 | 8.4 |
| *GDAS-CIFAR10 [17] | 26.0 | 8.5 |
| *DSNAS-ImageNet [29] | 25.7 | 8.1 |
| *AutoFormer [8] | 25.3 | 7.4 |
| Sampling [24] | 25.3 | - |
| *SDARTS-ADV-CIFAR10 [9] | 25.2 | 7.8 |
| *PC-DARTS-CIFAR10 [66] | 25.1 | 7.8 |
| *ProxylessNAS-ImageNet [5] | 24.9 | 7.5 |
| *FairDARTS-ImageNet [13] | 24.4 | 7.4 |
| *PR-DARTS [73] | 24.1 | 7.3 |
| *DARTS$^-$-ImageNet [12] | 23.8 | 7.0 |
| *Darts2nd-cifar10 [41] | 26.7 | 8.7 |
| EC-darts2nd-cifar10 [51] | 26.4 | 8.5 |
| CDEP-darts2nd-cifar10 [50] | 26.5 | 8.5 |
| GMPGC-darts2nd-cifar10 [46] | 26.3 | 8.2 |
| SANAS-darts2nd-cifar10 (ours) | **24.8** | **8.3** |
| *Pdarts-cifar100 [11] | 24.7 | 7.5 |
| EC-pdarts-cifar100 [51] | 24.5 | 7.3 |
| CDEP-pdarts-cifar100 [50] | 24.6 | 7.4 |
| GMPGC-pdarts-cifar100 [46] | 24.5 | 7.4 |
| SANAS-pdarts-cifar100 (ours) | **23.8** | **6.6** |
| *Pcdarts-ImageNet [66] | 24.2 | 7.3 |
| EC-pcdarts-ImageNet [51] | 24.0 | 7.2 |
| CDEP-pcdarts-ImageNet [50] | 23.9 | 7.3 |
| GMPGC-pcdarts-ImageNet [46] | 24.0 | 7.3 |
| SANAS-pcdarts-ImageNet (ours) | **22.2** | **6.1** |

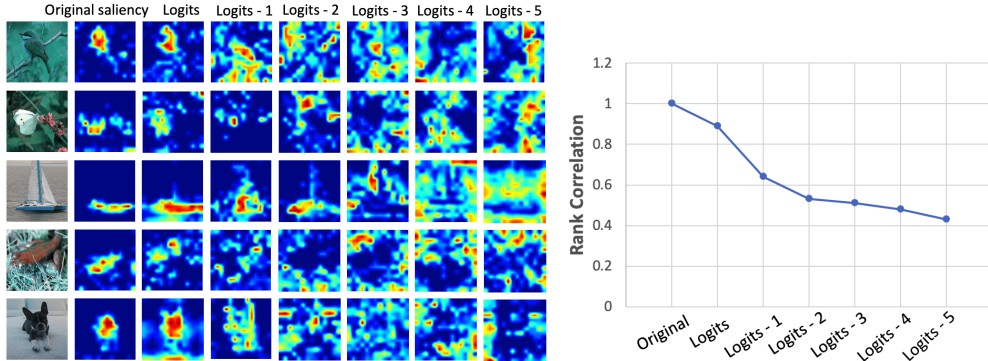

Figure 2: Sanity check of saliency maps. Logits−$n$ is the $n$-th layer below the logits layer.

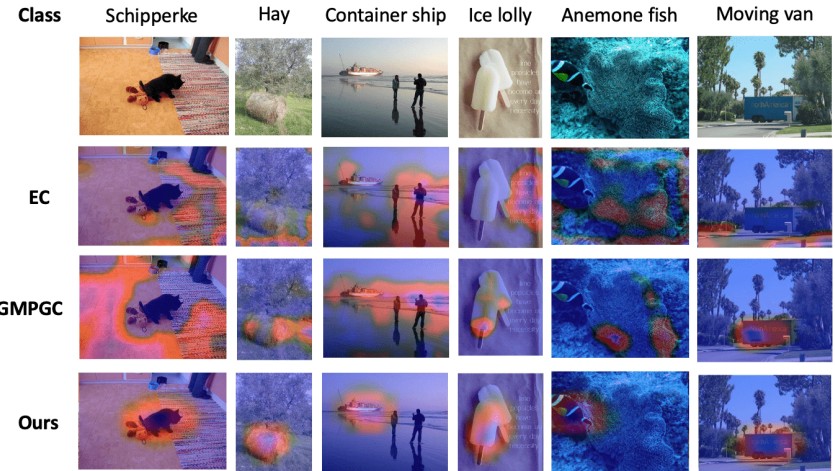

Figure 3: Visualization of saliency maps.

**Human evaluation of saliency** We randomly sampled 100 images from the test set of ImageNet and generated saliency maps for them using different methods. Then we asked three undergraduates to judge whether the saliency maps are sensible. The ratings are from 1-5 (higher is better). Table 3 summarizes the mean and standard deviation of ratings. Our methods achieve significantly higher ratings (significance test results are in the supplements), which demonstrates that our methods can generate more accurate saliency maps than the baselines. Inter-annotator Kappa score is 0.71, which shows strong agreements among annotators.

Table 3: Human evaluation of saliency.

|  | Ratings |
| --- | --- |
| Darts2nd | 3.30±0.24 |
| EC-darts2nd [51] | 3.42±0.16 |
| CDEP-darts2nd [50] | 3.58±0.11 |
| GMPGC-darts2nd [46] | 3.51±0.12 |
| SANAS-darts2nd (ours) | **3.93**±0.06 |
| Pdarts | 3.26±0.14 |
| EC-pdarts [51] | 3.59±0.21 |
| CDEP-pdarts [50] | 3.46±0.19 |
| GMPGC-pdarts [46] | 3.50±0.09 |
| SANAS-pdarts (ours) | **4.07**±0.11 |

**Visualization of saliency** In Figure 3, we visualize the saliency maps generated for some randomly-sampled ImageNet images. These saliency maps are very sensible. Warmer color (representing higher saliency) regions correspond to objects. Colder color regions correspond to background. These results show that our method is effective in generating correct saliency maps. In contrast, the saliency maps generated by baselines are less sensible. For example, in the schipperke, hay, container ship, ice lolly images, higher saliency regions of baselines are in the background instead of on objects.

**Ablation studies** In terms of how to use saliency scores to reweight pixels, we compare the element-wise product between pixels and saliency scores in Eq.(3) with 1) element-wise addition between pixels and saliency scores; 2) element-wise product between pixels and the absolute values of saliency

Table 4: Test errors of different reweighting mechanisms.

| | CIFAR-100 | | CIFAR-10 | |
|---|---|---|---|---|
| | Ours+darts | Ours+pdarts | Ours+darts | Ours+pdarts |
| Product | **16.4**±0.09 | **15.2**±0.09 | **2.54**±0.05 | **2.45**±0.03 |
| Addition | 20.3±0.27 | 17.7±0.03 | 2.71±0.11 | 2.53±0.06 |
| Absolute | 16.8±0.11 | 15.4±0.12 | 2.55±0.08 | 2.48±0.07 |
| Concatenate | 17.9±0.11 | 16.4±0.06 | 2.74±0.07 | 2.61±0.08 |
| No reweight | 20.6±0.44 | 17.5±0.06 | 2.76±0.09 | 2.54±0.04 |

scores; 3) concatenate saliency map with input image and feed the concatenation into the second model $W_2$; 4) no reweighting. Table 4 shows results where our method is applied to DARTS2nd and P-DARTS. From this table, we make the following observations. **First**, product works better than addition. The reason is: magnitude of saliency scores (perturbations) is very small; adding them to pixels does not render a significant change of pixel values, and consequently cannot distinguish important pixels from unimportant ones. In contrast, the relative difference between saliency scores is significantly large; multiplying them to pixels can better distinguish important and unimportant pixels. **Second**, reweighting pixels using signed saliency score and absolute saliency scores does not have a significant difference. This shows that signs of saliency scores do not matter too much. **Third**, product works better than concatenation. The possible reason is: compared with concatenation, product can better differentiate important and unimportant pixels using the saliency scores. **Fourth**, reweighting pixels works better than no reweighting. This demonstrates that multiplying saliency scores to inputs is indeed helpful in identifying important pixels, which helps to improve classification performance.

In the next ablation study, we compare the adversarial attack (AA) based saliency detection method with another two saliency detection methods, including integrated gradients (IG) [59] and SmoothGrad (SG) [56] by plugging them into our framework. These studies are performed on Darts2nd and Pdarts. Table 5 shows the results, where we make two observations. First, our framework with IG and SG still outperforms vanilla Darts2nd and Pdarts. This demonstrates that our framework is a general one that generalizes beyond a single saliency detection method. Second, IG and SG perform worse than AA. A possible reason is: IG and SG restrict the definition of saliency to be gradient-based. In contrast, AA treats saliency scores as optimization variables and automatically learns them by solving an optimization problem, which is more flexible.

To better understand the effectiveness of the proposed four stages performed end-to-end, we compare with the following two ablation settings, which are performed on Darts2nd and Pdarts.

- Perform the four stages separately (denoted as **Separate**) instead of end-to-end.

- Perform stages I, II, III by optimizing the weighted sum of their objective functions with weights 1, 0.5, 1, in a multi-task learning (MTL) manner (denoted as **MTL**).

Table 7 shows the results on Separate and MTL. We make two observations. First, our end-to-end method works better than Separate which conducts the four stages separately. Conducting the four stages end-to-end can enable them to mutually influence each other to achieve the overall best performance. In contrast, when conducted separately, earlier stages cannot be influenced by later stages (e.g., stage I cannot be influenced by stage IV), which leads to worse performance. Second, our method performs better than MTL. The tasks in stages I-III have an inherent order: before detecting saliency maps using a model, we first need to train this model; before training the second model on saliency-reweighted data, we need to detect the saliency maps first. MTL performs these three tasks simultaneously by minimizing a single objective, which breaks their inherent order and therefore leads to worse performance. In contrast, our method preserves this order using multi-level optimization.

Table 5: Ablation results on saliency detection methods.

| | CIFAR-100 | CIFAR-10 |
|---|---|---|
| Darts [41] | 20.58±0.44 | 2.76±0.09 |
| IG+darts | 16.92±0.08 | 2.62±0.06 |
| SG+darts | 17.05±0.11 | 2.59±0.03 |
| AA+darts | **16.42**±0.09 | **2.54**±0.05 |
| Pdarts [11] | 17.52±0.06 | 2.54±0.04 |
| IG+pdarts | 15.83±0.08 | 2.47±0.03 |
| SG+pdarts | 15.81±0.05 | 2.48±0.04 |
| AA+pdarts | **15.16**±0.09 | **2.45**±0.03 |

Table 6: Results on the GLUE benchmark. "# Param." denotes the number of model parameters. "Infer" denotes the speedup of inference time compared with $BERT_{12}$.

| Method | # Param. | Infer | SST-2 | MRPC | QQP | MNLI | QNLI | RTE | Average |
|---|---|---|---|---|---|---|---|---|---|
| $BERT_{12}$ | 109M | 1x | 93.5 | 88.9 | 71.2 | 84.6 | 90.5 | 66.4 | 82.5 |
| $BERT_{12}$-T | 109M | 1x | 93.3 | 88.7 | 71.1 | 84.8 | 90.4 | 66.1 | 82.4 |
| $BERT_6$-PKD | 67.0M | 1.9x | 92.0 | 85.0 | 70.7 | 81.5 | 89.0 | 65.5 | 80.6 |
| $BERT_3$-PKD | 45.7M | 3.7x | 87.5 | 80.7 | 68.1 | 76.7 | 84.7 | 58.2 | 76.0 |
| $DistilBERT_4$ | 52.2M | 3.0x | 91.4 | 82.4 | 68.5 | 78.9 | 85.2 | 54.1 | 76.8 |
| $TinyBert_4$ | 14.5M | 9.4x | 92.6 | 86.4 | 71.3 | 82.5 | 87.7 | 62.9 | 80.6 |
| $BiLSTM_{SOFT}$ | 10.1M | 7.6x | 90.7 | - | 68.2 | 73.0 | - | - | - |
| AdaBERT | 8.3M | 16.1x | 91.9 | 85.3 | 70.2 | 81.9 | 86.9 | 64.8 | 80.2 |
| EC-AdaBERT | 8.7M | 15.8x | 91.9 | 85.6 | 70.7 | 81.8 | 86.9 | 65.0 | 80.3 |
| CDEP-AdaBERT | 8.8M | 16.4x | 92.5 | 85.9 | 70.6 | 82.2 | 87.4 | 65.2 | 80.6 |
| GMPGC-AdaBERT | 8.1M | 15.5x | 92.0 | 85.6 | 71.4 | 82.8 | 87.1 | 65.0 | 80.7 |
| Ours-AdaBERT | 8.2M | 16.3x | 93.4 | 87.0 | 71.8 | 83.7 | 88.5 | 66.6 | 81.8 |

Figure 4(right) shows how the test error of SANAS-darts2nd on CIFAR100 varies with $\gamma$. When $\gamma = 0$, the validation loss of $W_1$ is not used for architecture search and the performance is inferior (compared with $\gamma = 2$). A $\gamma$ in the middle ground which properly balances the two validation losses yields the optimal performance. Using the validation loss of $W_1$ only is equivalent to vanilla DARTS-2nd (results are in the supplements).

## 4.2 Experiments on text classification

In this section, we apply the proposed framework for text classification. The Gumbel softmax trick [31] is leveraged to deal with non differentiability of texts.

**Dataset** We conduct experiments on six datasets in the GLUE benchmark [63]: SST-2, MRPC, QQP, MNLI, QNLI and RTE. SST-2 is a sentiment classification dataset where the input text is movie review and the output label is whether the review is positive or negative. In MRPC and QQP, the input is a pair of sentences and the output is whether they are semantically equivalent. MNLI, QNLI, and RTE are textual entailment recognition datasets.

**Baselines** We compare with the following baselines: 1) BERT [16], 2) BERT-PKD [58], 3) Distil-BERT [52], 4) TinyBERT [33], 5) $BiLSTM_{SOFT}$ [62], 6) AdaBERT [7], 7) EC-AdaBERT [51], 8) CDEP-AdaBERT [50], and 9) GMPGC-AdaBERT [46].

**Hyperparameter settings** Candidate operations are commonly used operations in convolutional networks, including 1D convolution, dilated convolution, pooling, identify, and zero. In dilated convolution, the kernel size includes 3, 5, and 7. Each convolution operation consists of an Relu-Conv-BatchNorm sequence. For pooling, we used average pooling and max pooling, where the kernel size is set to 3. The "SAME" padding is utilized for convolution and pooling. We optimize weight parameters using SGD. The initial learning rate is set to $2e - 2$. It is annealed

Table 7: Ablation results on Separate and MTL.

| | CIFAR-100 | CIFAR-10 |
|---|---|---|
| Separate+darts | 18.05±0.27 | 2.68±0.06 |
| MTL+darts | 18.26±0.12 | 2.70±0.05 |
| Ours+darts | **16.42**±0.09 | **2.54**±0.05 |
| Separate+pdarts | 16.49±0.07 | 2.51±0.03 |
| MTL+pdarts | 16.83±0.10 | 2.52±0.04 |
| Ours+pdarts | **15.16**±0.09 | **2.45**±0.03 |

using a cosine scheduler. The momentum is set to 0.9. We use Adam [34] to optimize the architecture variables. The learning rate is set to $3e - 4$ and weight decay is set to $1e - 3$.

**Main results** Table 6 shows the results. We make the following observations. First, our method works better than AdaBERT, which is an NAS method without saliency detection. This further demonstrates the effectiveness of saliency detection in improving NAS. Second, our method works better than EC, CDEP, and GMPGC. This further shows that performing saliency detection and NAS jointly is better than conducting them separately as the three baselines do. Third, compared

with $BERT_{12}$ and $BERT_{12}$-T, our method achieves similar performance while using much fewer parameters and being much faster during inference.

**Qualitative results**   Table 8 shows salient words detected by different methods on a randomly sampled sentence (whose sentiment is labeled as being positive). As can be seen, our method can successfully recognize the words "entertaining" and "please" which are mostly relevant to a positive sentiment. In contrast, the two baselines fail to do that.

## 5   Conclusions and discussions

In this paper, we propose to leverage the saliency information of input data to improve NAS. Our work makes the following contributions. First, our method can detect saliency and perform NAS end-to-end, based on a four-level optimization framework. The framework performs four stages in a unified way: train a preliminary model, generate saliency maps using the preliminary model, retrain the model on saliency-reweighted data, and update architecture by minimizing validation losses. Second, our

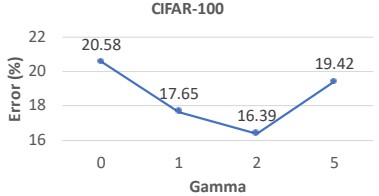

Figure 4: How errors change with $\gamma$.

framework is end-to-end differentiable, allowing using efficient gradient-based algorithms as solvers. Third, our method provides a mechanism to evaluate generated saliency maps by checking whether they are helpful for improving classification performance. We demonstrate the effectiveness of our method on several datasets.

One major limitation of this work is that it cannot be easily applied to non-differentiable NAS methods that are based on reinforcement learning (RL) and evolutionary algorithm (EA). The reason is that our method uses a gradient-based optimization algorithm to solve the multi-level optimization problem. For non-differentiable NAS methods, their non-differentiable objective functions do not have gradients, which therefore are not compatible with the gradient-based algorithm used by our method. Please see Appendix **??** for discussion on how to extend our method to non-differentiable NAS methods. Another limitation of our method is its higher time cost than baselines, due to the extra computation needed for detecting saliency maps. Considering the benefits and limitations of our method, we recommend using our method in applications that strongly need high-performance architectures capable of generating sensible saliency maps but do not have strong efficiency requirements on architecture search time. For applications which have high restrictions on search cost but allow sacrificing some performance and ignor-

Table 8: Top-2 salient words (marked with red color) detected by different methods.

| EC | an entertaining *mix* of period *drama* and flat-out farce that should please history fans. |
|---|---|
| GMPGC | an *entertaining* mix of period drama and flat-out farce that should please *history* fans. |
| Ours | an *entertaining* mix of period drama and flat-out farce that should *please* history fans. |
| EC | a very witty take on change, *risk* and romance, and the film *uses* humor to make its points about acceptance and growth. |
| GMPGC | a very witty take on change, risk and romance, and the *film* uses humor to make its points about *acceptance* and growth. |
| Ours | a very *witty* take on change, risk and romance, and the film uses *humor* to make its points about acceptance and growth. |

ing saliency maps, other NAS methods might be better choices. Please see Appendix **??** for a more detailed discussion.

One potential negative societal impact of our work is: in mission-critical applications such as disease diagnosis and autonomous driving, if saliency maps generated by our method are not correct, they may mislead human decision-makers. For future works, we plan to investigate these ideas: 1) formulate saliency-based network pruning [74] as a saliency-aware NAS problem and automatically search for the optimal pruning decisions based on detected saliency; 2) extend the notion of saliency from input data to blocks in neural networks, develop multi-level optimization based frameworks to detect the saliency of blocks, and perform pruning on blocks based on detected saliency.

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
