# Supplementary Materials
## – Saliency-Aware Neural Architecture Search

## Abstract

In this supplement, we present the optimization algorithm, details of experimental settings, and additional experimental results.

## Contents

## Appendix A. Appendix

### A.1. More discussion of limitations

**Apply SANAS to non-differentiable NAS methods.** To apply SANAS to non-differentiable NAS methods, we have to change the current gradient-based optimization algorithm to some other non-gradient-based optimization algorithms (such as REINFORCE (Williams, 1992) for reinforcement learning), which might incur higher computational costs. To apply SANAS to reinforcement learning (RL) based NAS methods, we perform the following procedures. First, we use an RL controller (Zoph and Le, 2017) to generate a set of candidate architectures. Second, given a candidate architecture, we train its weight parameters on a training dataset, similar to stage I in SANAS. Third, given the trained model, we perform adversarial attacks to detect the saliency maps of the training data, similar to stage II in SANAS. Fourth, we use saliency maps to reweight training data and retrain the model on reweighted data, similar to stage III in SANAS. Fifth, we evaluate the retrained model on a validation set and use validation accuracy as a reward for this architecture. We repeat steps 2-5 for every candidate architecture, calculate the mean reward on all candidate architectures, and update the RL controller by maximizing the mean reward using policy gradient (Zoph and Le, 2017). These procedures repeat until convergence. Similar procedures can be conducted to perform saliency-aware architecture search in evolutionary algorithm based NAS methods.

**When to use our method and when not.** It is recommended to use SANAS in applications that strongly need high-performance neural architectures capable of generating sensible saliency maps but do not have strong requirements on the time spent on architecture search. For example, imaging-based disease diagnosis is a good application scenario of SANAS, for two reasons. First, disease diagnosis needs to be highly accurate and needs high-fidelity saliency maps for interpreting predicted diagnosis outcomes. Second, to use an automatically searched neural architecture in hospitals, FDA approval is needed, which usually takes several months. To successfully pass the FDA approval, it is acceptable to take some extra time to search for a high-quality neural architecture. For applications which have high restrictions on search cost but allow sacrificing some performance and ignoring saliency maps, other NAS methods that have higher search efficiency but lower performance and weaker saliency-generation capability than our method might be better choices. Examples of such applications are online learning applications which need to update architectures in real time.

### A.2. More examples of visual saliency maps

Figure 1 shows more examples of visual saliency maps. The saliency maps detected by our method are more sensible than those detected by baselines.

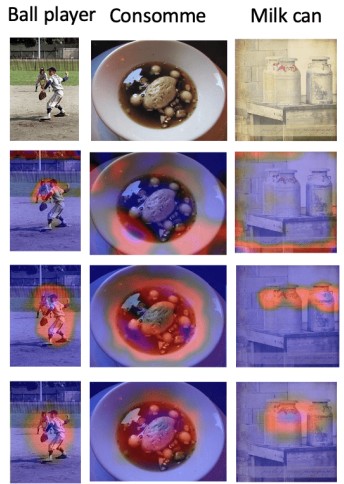

Figure 1: More examples of visual saliency maps.

Table 1: Top-2 salient words (marked with red color) detected by different methods.

| EC | with youthful high spirits, tautou remains captivating throughout michele's religious and romantic quests, and she is backed by a likable cast. |
|---|---|
| GMPGC | with youthful high spirits, tautou remains captivating throughout michele's religious and romantic quests, and she is backed by a likable cast. |
| Ours | with youthful high spirits, tautou remains captivating throughout michele's religious and romantic quests, and she is backed by a likable cast. |
| EC | the title's lameness should clue you in on how bad the movie is. |
| GMPGC | the title's lameness should clue you in on how bad the movie is. |
| Ours | the title's lameness should clue you in on how bad the movie is. |

## A.3. More examples of salient word detection

Table 1 shows more examples of salient words detected by different methods. In each example, the top-2 words detected by our method are more salient than baselines. The prediction task corresponding to Table 8 in the main paper and 1 is sentiment classification. A word is more salient if it has a stronger correlation with a sentiment (either positive or negative). For example, the word "entertaining" in Table 8 in the main paper implies a positive sentiment, and therefore is considered to be salient. In contrast, the word "mix" is a neutral word that is irrelevant to sentiments, and therefore is not considered to be salient.

### A.4. Improve computational efficiency of SANAS

We improved the computational efficiency of our method from both the algorithm side and implementation side. On the algorithm side, we speed up computation by approximating the optimal solution at each stage using a one-step gradient descent update (Liu et al., 2019) and reducing the frequencies of these updates. Specifically, we update the architecture $A$ every 5 mini-batches. In contrast, baselines (including Darts, Parts, Pcdarts, Prdarts) update $A$ on every mini-batch. We update model weights $W_2$ and perturbations $\delta$ every 3 mini-batches, and update $W_1$ on every mini-batch. We empirically found that reducing the update frequencies of certain parameters can significantly speed up convergence without sacrificing performance. Besides, when calculating hypergradients of $A$, we recursively approximate matrix-vector multiplications using finite-difference calculations (Liu et al., 2019), which reduces the computation cost from being quadratic in matrix dimensions down to linear.

On the implementation side, we speed up computation by leveraging techniques and tricks including 1) automatic mixed precision (Micikevicius et al., 2017), 2) using multiple (4, specifically) workers and pinned memory in PyTorch DataLoader, 3) using cudNN autotuner, 4) kernel fusion, etc.

### A.5. Hyperparameter tuning strategies

Most hyperparameters in our method follow their default values used in baseline methods. The only hyperparameter needing to be tuned is the tradeoff parameter $\gamma$. To tune $\gamma$ on CIFAR-100, we randomly sample 2.5K data from the 25K training set and sample 2.5K data from the 25K validation set. Then we use the 5K sampled data as a hyperparameter tuning set. $\gamma$ is tuned in 0.1, 0.5, 1, 2, 3. For each configuration of $\gamma$, we use the remaining 22.5K training data and 22.5K validation data to perform architecture search and use their combination to perform architecture evaluation (retraining a larger stacked network from scratch). Then we measure the performance of the stacked network on the 5K sampled data. $\gamma$ value yielding the best performance on the 5K sampled data is selected. For $\gamma$ in CIFAR-10 and ImageNet experiments, we simply used the value tuned on CIFAR-100 and did not conduct further tuning.

## Appendix B. Optimization Algorithm

We use a well-established algorithm developed in (Liu et al., 2019) to solve the proposed SANAS problem. Theoretic convergence of this algorithm has been broadly analyzed in (Ghadimi and Wang, 2018; Grazzi et al., 2020; Ji et al., 2021; Liu et al., 2021; Yang et al., 2021). At each level of optimization problem, the optimal solution (on the left-hand side of the equal sign, marked with $^*$), its exact value is computationally expensive to compute. To address this problem, following (Liu et al., 2019), we approximate the optimal solution using a one-step gradient descent update and plug the approximation into the next level of optimization problem. In the sequel, $\frac{\partial \cdot}{\partial \cdot}$ denotes partial derivative. $\frac{d \cdot}{d \cdot}$ denotes an ordinary derivative. $\nabla^2_{Y,X} f(X, Y)$ denotes $\frac{\partial f(X,Y)}{\partial X \partial Y}$.

First of all, we approximate $W_1^*(A)$ using

$$W_1' = W_1 - \xi_{w_1} \nabla_{W_1} L(W_1, A, D^{(\text{tr})}) \tag{1}$$

where $\xi_{w_1}$ is a learning rate. Plugging $W_1'$ into $\ell(f(x_i + \delta_i; W_1^*(A), A), f(x_i; W_1^*(A), A))$, we obtain an approximated objective $O_{\delta_i} = \ell(f(x_i + \delta_i; W_1', A), f(x_i; W_1', A))$. Then we approximate $\delta_i^*(W_1^*(A), A)$ using one-step gradient ascent update of $\delta_i$ with respect to $O_{\delta_i}$:

$$\delta_i' = \delta_i + \xi_\delta \nabla_{\delta_i} \ell(f(x_i + \delta_i; W_1', A), f(x_i; W_1', A)). \tag{2}$$

Plugging $\delta_i'$ into $\sum_{i=1}^{N} \ell(f(\delta_i^*(W_1^*(A), A) \odot x_i; W_2, A), t_i)$, we obtain an approximated objective $O_{W_2} = \sum_{i=1}^{N} \ell(f(\delta_i' \odot x_i; W_2, A), t_i)$. Then we approximate $W_2^*(\{\delta_i^*(W_1^*(A), A)\}_{i=1}^{N}, A)$ using one-step gradient descent update of $W_2$ with respect to $O_{W_2}$:

$$W_2' = W_2 - \xi_{w_2} \nabla_{W_2} (\sum_{i=1}^{N} \ell(f(\delta_i' \odot x_i; W_2, A), t_i)). \tag{3}$$

Finally, we plug $W_1'$ and $W_2'$ into $L(W_2^*(\{\delta_i^*(W_1^*(A), A)\}_{i=1}^{N}, A), A, D^{(\text{val})}) + \gamma L(W_1^*(A), A, D^{(\text{val})})$ and get $O_A = L(W_2', A, D^{(\text{val})}) + \gamma L(W_1', A, D^{(\text{val})})$. We can update the architecture $A$ by descending the gradient of $O_A$ w.r.t $A$:

$$A \leftarrow A - \eta(\nabla_A L(W_2', A, D^{(\text{val})}) + \gamma L(W_1', A, D^{(\text{val})}))) \tag{4}$$

where

$$\begin{aligned} \nabla_A L(W_1', A, D^{(\text{val})}) = \\ \frac{dW_1'}{dA} \frac{\partial L(W_1', A, D^{(\text{val})})}{\partial W_1'} + \frac{\partial L(W_1', A, D^{(\text{val})})}{\partial A} = \\ -\xi_{w_1} \nabla_{A, W_1}^2 L(W_1, A, D^{(\text{tr})}) \nabla_{W_1'} L(W_1', A, D^{(\text{val})}) + \frac{\partial L(W_1', A, D^{(\text{val})})}{\partial A} \end{aligned} \tag{5}$$

The first term in the third line involves expensive matrix-vector product, whose computational complexity can be reduced by a finite difference approximation:

$$\nabla_{A, W_1}^2 L(W_1, A, D^{(\text{tr})}) \nabla_{W_1'} L(W_1', A, D^{(\text{val})}) \approx \frac{1}{2\alpha} (\nabla_A L(W_1^+, A, D^{(\text{tr})}) - \nabla_A L(W_1^-, A, D^{(\text{tr})})), \tag{6}$$

where $W_1^{\pm} = W_1 \pm \alpha \nabla_{W_1'} L(W_1', A, D^{(\text{val})})$ and $\alpha$ is a small scalar that equals $0.01/\|\nabla_{W_1'} L(W_1', A, D^{(\text{val})})\|_2$. Let $\Delta'$ denote $\{\delta_i'\}_{i=1}^{N}$. For $\nabla_A L(W_2', A, D^{(\text{val})})$ in Eq.(4), it can be calculated as

$$\frac{\partial W_2'}{\partial A} \frac{\partial L(W_2', A, D^{(\text{val})})}{\partial W_2'} + \frac{\partial L(W_2', A, D^{(\text{val})})}{\partial A} \tag{7}$$

where

$$\frac{\partial W_2'}{\partial A} = \frac{\partial \Delta'}{\partial A} \frac{\partial W_2'}{\partial \Delta'} + \frac{\partial W_2'}{\partial A} \tag{8}$$

$$\frac{\partial \Delta'}{\partial A} = \frac{dW_1'}{dA} \frac{\partial \Delta'}{\partial W_1'} + \frac{\partial \Delta'}{\partial A} \tag{9}$$

according to the chain rule, where

$$\frac{\partial W_2'}{\partial \Delta'} = \frac{\partial(W_2 - \xi_{w_2} \nabla_{W_2}(\sum_{i=1}^{N} \ell(f(\delta_i' \odot x_i; W_2, A), t_i)))}{\partial \Delta'} \tag{10}$$

$$= -\xi_{w_2} \nabla_{\Delta', W_2}^2 (\sum_{i=1}^{N} \ell(f(\delta_i' \odot x_i; W_2, A), t_i)), \tag{11}$$

$$\frac{\partial W_2'}{\partial A} = \frac{\partial(W_2 - \xi_{w_2}\nabla_{W_2}(\sum_{i=1}^{N}\ell(f(\delta_i' \odot x_i; W_2, A), t_i)))}{\partial A} \tag{12}$$

$$= -\xi_{w_2}\nabla_{A,W_2}^2(\sum_{i=1}^{N}\ell(f(\delta_i' \odot x_i; W_2, A), t_i)) \tag{13}$$

$$\frac{\partial \Delta'}{\partial W_1'} = \frac{\partial(\Delta + \xi_{\delta}\nabla_{\Delta}(\sum_{i=1}^{N}\ell(f(x_i + \delta_i; W_1', A), f(x_i; W_1', A))))}{\partial W_1'} \tag{14}$$

$$= \xi_{\delta}\nabla_{W_1',\Delta}^2(\sum_{i=1}^{N}\ell(f(x_i + \delta_i; W_1', A), f(x_i; W_1', A))), \tag{15}$$

$$\frac{\partial \Delta'}{\partial A} = \frac{\partial(\Delta + \xi_{\delta}\nabla_{\Delta}(\sum_{i=1}^{N}\ell(f(x_i + \delta_i; W_1', A), f(x_i; W_1', A))))}{\partial A} \tag{16}$$

$$= \xi_{\delta}\nabla_{A,\Delta}^2(\sum_{i=1}^{N}\ell(f(x_i + \delta_i; W_1', A), f(x_i; W_1', A))), \tag{17}$$

$$\frac{dW_1'}{dA} = \frac{d(W_1 - \xi_{w_1}\nabla_{W_1}L(W_1, A, D^{(\text{tr})}))}{dA} \tag{18}$$

$$= -\xi_{w_1}\nabla_{A,W_1}^2 L(W_1, A, D_t^{(\text{tr})}). \tag{19}$$

The gradient descent update of $A$ in equation 4 can run one or more steps. After $A$ is updated, the one-step gradient-descent approximations (in equation 1-3), which are functions of $A$, change with $A$ and need to be re-updated. Then, the gradient of $A$, which is a function of one-step gradient-descent approximations, needs to be re-calculated and is used to refresh $A$. In sum, the update of $A$ and the updates of one-step gradient-descent approximations mutually depend on each other. These updates are performed iteratively until convergence. This algorithm is summarized in Algorithm 1.

---

**Algorithm 1** Optimization algorithm for SANAS

---

**while** *not converged* **do**

   1. Update the approximation $W_1'$ of $W_1^*(A)$ using equation 1
   2. Update the approximation $\{\delta_i'\}_{i=1}^{N}$ of $\{\delta_i\}_{i=1}^{N}$ using equation 2
   2. Update the approximation $W_2'$ of $W_2^*(\{\delta_i^*(W_1^*(A), A)\}_{i=1}^{N}, A)$ using equation 3
   3. Update the architecture $A$ using equation 4

**end**

---

In the gradient of $A$ calculated using chain rule, the number of chains is the same as the number of levels in our proposed four-level optimization formulation. This shows that this optimization algorithm preserves the four-level nested optimization nature of the proposed SANAS formulation.

## Appendix C. Results on inference costs

To perform inference, we first calculate saliency map, then do prediction on saliency-reweighted image. To reduce inference cost, we can reduce the number of layers in the searched architecture. We performed such an experiment. Table 2 shows that while having similar inference costs as baselines, our method achieves significantly lower test errors.

Table 2: Test errors on Cifar100/10 (C100/10), inference time (ms)

| Method | Error-C100 | Error-C10 | Infer time (ms) |
|---|---|---|---|
| Darts2nd | 20.58±0.44 | 2.76±0.09 | 28.4 |
| EC-darts2nd | 20.05±0.31 | 2.83±0.12 | 29.6 |
| CDEP-darts2nd | 19.53±0.46 | 2.75±0.05 | 30.2 |
| GMPGC-darts2nd | 19.08±0.36 | 2.81±0.07 | 32.7 |
| Ours-darts2nd | **16.98**±0.12 | **2.60**±0.08 | 28.1 |

## Appendix D. Model parameters, search costs, and FLOPs on ImageNet

Table 3 shows the number of model parameters, search costs, and FLOPs on ImageNet. The parameter numbers, search costs, and FLOPs of our methods are close to those in differentiable baselines.

## Appendix E. Additional experimental results on CIFAR-10

Table 4 shows additional experimental results on CIFAR-10.

## Appendix F. Additional ablation study results

We experimented with training the model $W_1$ in the 1st stage and performing adversarial attack simultaneously, which leads to worse performance. For example, under the search space of Darts2nd, errors on CIFAR-100 and CIFAR-10 increased by 1.25% and 0.17% (absolute). The possible reason is: performing these two tasks simultaneously will make $W_1$ robust to small perturbations, which makes it difficult to sensitively detect saliency maps.

## Appendix G. Instructions given to participants in human studies

Figure 2 shows the screenshot of instructions given to participants in human studies.

## Appendix H. Additional discussions

The proposed saliency-aware method can be potentially applied to non-NAS methods including resnet and densenet, by replacing architecture variable $A$ in the first and fourth stage with some continuous hyperparameters of resnet or densenet.

Table 3: Top-1 and top-5 classification errors on the test set of ImageNet, number of model parameters (millions), search cost (GPU days), and FLOPs (M). Results marked with * are obtained from DARTS⁻ (Chu et al., 2020a) and DrNAS (Chen et al., 2020). The rest notations are the same as those in Table 1 in the main paper. From top to bottom, on the first three blocks are 1) networks manually designed by humans; 2) non-gradient based NAS methods; and 3) gradient-based NAS methods.

| Method | Top-1 Error (%) | Top-5 Error (%) | Param (M) | Cost (GPU days) | FLOPs (M) |
|---|---|---|---|---|---|
| *Inception-v1 (Szegedy et al., 2015) | 30.2 | 10.1 | 6.6 | - | 1448 |
| *MobileNet (Howard et al., 2017) | 29.4 | 10.5 | 4.2 | - | 569 |
| *ShuffleNet 2× (v1) (Zhang et al., 2018) | 26.4 | 10.2 | 5.4 | - | 524 |
| *ShuffleNet 2× (v2) (Ma et al., 2018) | 25.1 | 7.6 | 7.4 | - | 299 |
| *NASNet-A (Zoph et al., 2018) | 26.0 | 8.4 | 5.3 | 1800 | 564 |
| *PNAS (Liu et al., 2018) | 25.8 | 8.1 | 5.1 | 225 | 588 |
| *MnasNet-92 (Tan et al., 2019) | 25.2 | 8.0 | 4.4 | 1667 | 388 |
| *AmoebaNet-C (Real et al., 2019) | 24.3 | 7.6 | 6.4 | 3150 | 570 |
| *SNAS-CIFAR10 (Xie et al., 2019) | 27.3 | 9.2 | 4.3 | 1.5 | 522 |
| *BayesNAS-CIFAR10 (Zhou et al., 2019) | 26.5 | 8.9 | 3.9 | 0.2 | - |
| *PARSEC-CIFAR10 (Casale et al., 2019) | 26.0 | 8.4 | 5.6 | 1.0 | - |
| *GDAS-CIFAR10 (Dong and Yang, 2019) | 26.0 | 8.5 | 5.3 | 0.2 | 581 |
| *DSNAS-ImageNet (Hu et al., 2020) | 25.7 | 8.1 | - | - | 324 |
| *SDARTS-ADV-CIFAR10 (Chen and Hsieh, 2020) | 25.2 | 7.8 | 5.4 | 1.3 | - |
| *PC-DARTS-CIFAR10 (Xu et al., 2020) | 25.1 | 7.8 | 5.3 | 0.1 | 586 |
| *ProxylessNAS-ImageNet (Cai et al., 2019) | 24.9 | 7.5 | 7.1 | 8.3 | 465 |
| *FairDARTS-CIFAR10 (Chu et al., 2019) | 24.9 | 7.5 | 4.8 | 0.4 | 386 |
| *FairDARTS-ImageNet (Chu et al., 2019) | 24.4 | 7.4 | 4.3 | 3.0 | 440 |
| *DrNAS-ImageNet (Chen et al., 2020) | 24.2 | 7.3 | 5.2 | 3.9 | - |
| *DARTS⁺-ImageNet (Liang et al., 2019) | 23.9 | 7.4 | 5.1 | 6.8 | 582 |
| *DARTS⁻-ImageNet (Chu et al., 2020a) | 23.8 | 7.0 | 4.9 | 4.5 | 467 |
| *DARTS⁺-CIFAR100 (Liang et al., 2019) | 23.7 | 7.2 | 5.1 | 0.2 | 591 |
| *DARTS2nd-CIFAR10 (Liu et al., 2019) | 26.7 | 8.7 | 4.7 | 4.0 | 574 |
| Ours-DARTS2nd-CIFAR10 | **24.9** | **8.4** | 4.9 | 4.6 | 526 |
| *PDARTS (CIFAR10) (Chen et al., 2019) | 24.4 | 7.4 | 4.9 | 0.3 | 557 |
| Ours-PDARTS-CIFAR10 | **23.9** | **6.8** | 4.9 | 1.1 | 517 |
| *PDARTS (CIFAR100) (Chen et al., 2019) | 24.7 | 7.5 | 5.1 | 0.3 | 577 |
| Ours-PDARTS-CIFAR100 | **23.8** | **6.7** | 5.2 | 1.1 | 535 |
| *PCDARTS-ImageNet (Xu et al., 2020) | 24.2 | 7.3 | 5.3 | 3.8 | 597 |
| Ours-PCDARTS-ImageNet | **22.4** | **6.2** | 5.4 | 4.1 | 540 |

# Appendix I. Experimental details of neural architecture search

### I.1. DARTS based experiments

For methods based on DARTS2nd, including Ours-darts2nd, GMPGC-darts2nd, CDEP-darts2nd, EC-darts2nd, the experimental settings are similar. In search spaces of DARTS, the candidate operations include: $3 \times 3$ and $5 \times 5$ separable convolutions, $3 \times 3$ and $5 \times 5$ dilated separable convolutions, $3 \times 3$ max pooling, $3 \times 3$ average pooling, identity, and zero. The network is a stack of multiple cells. The stride of all operations is set to 1.

Table 4: Test error on CIFAR-10, number of model weights (millions), and search cost (GPU days on a Tesla v100). Results marked with * are obtained from DARTS⁻ (Chu et al., 2020a), NoisyDARTS (Chu et al., 2020b), and DrNAS (Chen et al., 2020). The rest notations are the same as those in Table 1 of the main paper.

| Method | Error(%) | Param(M) | Cost |
|---|---|---|---|
| *DARTS-1st (Liu et al., 2019) | 3.00±0.14 | 3.3 | 0.4 |
| Ours-DARTS1st | **2.77**±0.02 | 3.4 | 0.8 |

In the given image, heatmap denotes saliency of regions. Warmer color denotes the region is more salient (e.g., corresponding to objects) and colder color denotes the region is less salient (e.g., corresponding to background).
Please give a rating for the heatmap regarding whether it is sensible. Specifically, whether the warmer color regions are truly salient and colder-color regions are not salient.
Please select a rating below.
5 – 80% of the heatmap is sensible.
4 – 60% of the heatmap is sensible.
3 – 40% of the heatmap is sensible.
2 – 20% of the heatmap is sensible.
1 – 0% of the heatmap is sensible.

Here are two examples.

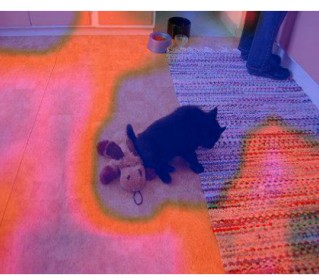 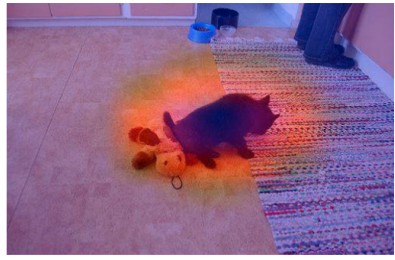

Rating: 1                    Rating: 5

Figure 2: Screenshot of instructions given to participants in human studies.

The convolved feature maps are padded to preserve their spatial resolution. The order for convolutional operations is ReLU-Conv-BN. Each separable convolution is applied twice. The convolutional cell has 7 nodes. The output node is the depthwise concatenation of all intermediate nodes, excluding the input nodes. The first and second nodes of cell $k$ are equal to the outputs of cell $k-2$ and cell $k-1$, respectively. $1\times1$ convolutions are inserted when necessary. Reduction cells are located at the 1/3 and 2/3 of the total depth of the network. In reduction cells, operations adjacent to the input nodes have a stride of 2.

For CIFAR-10 and CIFAR-100, during architecture search, the network is a stack of 8 cells, each consisting of 7 nodes, with the initial channel number set to 16. The search is performed for 50 epochs, with a batch size of 64. Network weights $W_1$ and $W_2$ were optimized using SGD with a learning rate of 0.025, a momentum of 0.9, and a weight decay of 0.0003. The architecture variables $A$ were optimized using Adam (Kingma and Ba, 2014) with a learning rate of 0.001, a momentum of $(0.5, 0.999)$, and a weight decay of 0.001. The learning rate was scheduled with cosine scheduling. The architecture variables were initialized with zero initialization.

During architecture evaluation, for CIFAR-10 and CIFAR-100, a larger network is formed by stacking 20 copies of the searched cell. The composed large network is trained on the combination of $D^{(\text{tr})}$ and $D^{(\text{val})}$. The initial channel number was set to 36. We trained the network with a batch size of 96, an epoch number of 600. On ImageNet, we evaluate two types of architectures: 1) those searched on a subset of ImageNet; 2) those searched on CIFAR-10 or CIFAR-100. In either type, 14 copies of optimally searched cells are stacked into a large network, which was trained on the 1.2M training images, with a batch size of 1024 and an epoch number of 250. Initial channel number was set to 48. Other hyperparameters are the same as those in architecture search. Cutout, path dropout of probability 0.2 and auxiliary towers with weight 0.4 were applied.

## I.2. PC-DARTS based experiments

For methods based on PC-DARTS, including Ours-pcdarts, GMPGC-pcdarts, CDEP-pcdarts, EC-pcdarts, the experimental settings are similar. The search space of PC-DARTS follows that of DARTS. For architecture search on CIFAR-100 and CIFAR-10, the hyperparameter $K$ was set to 4. The network is a stack of 8 cells. Each cell contains 6 nodes. Initial channel number is set to 16. The architecture variables are trained using the Adam optimizer for 50 epochs. The learning rate is set to $6e-4$, without decay. The weight decay is set to $1e-3$. The momentum is set to $(0.5, 0.999)$. The network weight parameters are trained using SGD for 50 epochs. The initial learning rate is set to 0.1. Cosine scheduling is used to decay the learning rate, down to 0 without restart. The momentum is set to 0.9. The weight decay is set to $3e-4$. The batch size is set to 256. Warm-up is utilized: in the first 15 epochs, architecture variables are frozen and only network weights are optimized.

The settings for architecture evaluation on CIFAR-100 and CIFAR-10 follow those of DARTS. 18 normal cells and 2 reduction cells are stacked into a large network. The initial channel number is set to 36. The stacked network is trained from scratch using SGD for 600 epochs, with batch size 128, initial learning rate 0.025, momentum 0.9, weight decay $3e-4$, norm gradient clipping 5, drop-path rate 0.3, and cutout. The learning rate is decayed to 0 using cosine scheduling without restart.

We combine our method and PC-DARTS to directly search for architectures on ImageNet. The stacked network starts with three convolution layers which reduce the input image resolution from 224×224 to 28×28, using stride 2. After the three convolution layers, 6 normal cells and 2 reduction cells are stacked. Each cell consists of $N = 6$ nodes. The sub-sampling rate was set to 0.5. The network was trained for 50 epochs. Architecture variables are trained using Adam. The learning rate is fixed to $6e-3$. The weight decay is set to $1e-3$. The momentum is set to $(0.5, 0.999)$. In the first 35 epochs, architecture

variables are frozen. Network weight parameters are trained using SGD. The initial learning rate is set to 0.5. It is decayed to 0 using cosine scheduling without restart. Momentum is set to 0.9. Weight decay is set to $3e - 5$. The batch-size is set to 1024. Epoch number is set to 250. Eight Tesla V100 GPUs were used.

For architecture evaluation on ImageNet, the stacked network starts with three convolution layers which reduce the input image resolution from 224×224 to 28×28, using stride 2. After the three convolution layers, 12 normal cells and 2 reduction cells are stacked. Initial channel number is set to 48. The network is trained from scratch using SGD for 250 epochs, with batch size 1024, initial learning rate 0.5, weight decay $3e - 5$, and momentum 0.9. For the first 5 epochs, learning rate warm-up is used. The learning rate is linearly decayed to 0. Label smoothing and auxiliary loss tower is used.

### I.3. P-DARTS based experiments

The search process has three stages. At the first stage, the search space and stacked network in P-DARTS are mostly the same as DARTS. The only difference is the number of cells in the stacked network in P-DARTS is set to 5. At the second stage, the number of cells in the stacked network is 11. At the third stage, the cell number is 17. At stage 1, 2, 3, the initial Dropout probability on skip-connect is 0, 0.4, and 0.7 for CIFAR-10, is 0.1, 0.2, and 0.3 for CIFAR-100; the size of operation space is 8, 5, 3, respectively. The final searched cell is limited to have 2 skip-connect operations at maximum. At each stage, the network is trained using the Adam optimizer for 25 epochs. The batch size is set to 96. The learning rate is set to 6e-4. Weight decay is set to 1e-3. Momentum is set to $(0.5, 0.999)$. In the first 10 epochs, architecture variables are frozen and only network weights are optimized.

For architecture evaluation on CIFAR-100 and CIFAR-10, the stacked network consists of 20 cells. The initial channel number is set to 36. The network is trained from scratch using SGD. The epoch number is set to 600. The batch size is set to 128. The initial learning rate is set to 0.025. The learning rate is decayed to 0 using cosine scheduling. Weight decay is set to 3e-4 for CIFAR-10 and 5e-4 for CIFAR-100. Momentum is set to 0.9. Drop-path probability is set to 0.3. Cutout regularization length is set to 16. Auxiliary towers of weight 0.4 are used.

For architecture evaluation on ImageNet, the settings are similar to those of DARTS. The network consists of 14 cells. The initial channel number is set to 48. The network is trained from scratch using SGD for 250 epochs. Batch size is set to 1024. Initial learning rate is set to 0.5. The learning rate is linearly decayed after each epoch. In the first 5 epochs, learning rate warmup is used. The momentum is set to 0.9. The weight decay is set to $3e - 5$. Label smoothing and auxiliary loss tower are used during training. The network was trained on 8 Nvidia Tesla V100 GPUs.

### I.4. PR-DARTS based experiments

The operations include: 3×3 and 5×5 separable convolutions, 3×3 and 5×5 dilated separable convolutions, 3×3 average pooling and 3×3 max pooling, zero, and skip connection. The stacked network consists of $k$ cells. The $k/3$- and $2k/3$-th cells are reduction cells. In reduction cells, all operations have a stride of two. The rest cells are normal cells. Operations in normal cells have a stride of one. Cells of the same type (either reduction or

normal) have the same architecture. The inputs of each cell are the outputs of two previous cells. Each cell contains four intermediate nodes and one output node. The output node is a concatenation of all intermediate nodes.

For architecture search on CIFAR-100 and CIFAR-10, the stacked network consists of 8 cells. The initial channel number is set to 16. In PR-DARTS, $\lambda_1$, $\lambda_2$, and $\lambda_3$ are set to 0.01, 0.005, and 0.005 respectively. The network was trained for 200 epochs. The mini-batch size is set to 128. Architecture variables are trained using Adam. The learning rate is set to $3e-4$. The weight decay is set to $1e-3$. Network weights are trained using SGD. The initial learning rate is set to 0.025. The momentum is set to 0.9. The weight decay is set to $3e-4$. The learning rate is decayed to 0 using cosine scheduling. For acceleration, per iteration, only two operations on each edge are randomly selected to update. The temperature $\tau$ is set to 10 and is linearly reduced to 0.1; $a = -0.1$ and $b = 1.1$. Pruning on each node is conducted by comparing the gate activation probabilities of all non-zero operations collected from all previous nodes and retaining top two operations.

For architecture evaluation on CIFAR10 and CIFAR100, the stacked network consists of 18 normal cells and 2 reduction cells. The initial channel number is set to 36. The network is trained from scratch using SGD. The mini-batch size is set to 128. The epoch number is set to 600. The initial learning rate is set to 0.025. The momentum is set to 0.9. The weight decay is set to $3e-4$. The gradient norm clipping is set to 5. The drop-path probability is set to 0.2. The cutout length is set to 16. The learning rate is decayed to 0 using cosine scheduling.

For architecture evaluation on ImageNet, the input images are resized to $224 \times 224$. The stacked network consists of 3 convolutional layers, 12 normal cells, and 2 reduction cells. The channel number is set to 48. The network is trained using SGD for 250 epochs. The batch size is set to 128. The learning rate is set to 0.025. The momentum is set to 0.9. The weight decay is set to $3e-4$. The gradient norm clipping is set to 5. The learning rate is decayed to 0 via cosine scheduling.

### I.5. Implementation details

Hyperparameters mostly follow those in DARTS (Liu et al., 2019), P-DARTS (Chen et al., 2019), PC-DARTS (Xu et al., 2020), and PR-DARTS (Zhou et al., 2020).

We use PyTorch to implement all models. The version of Torch is 1.4.0 (or above). We build our method upon official python packages for different differentiable search approaches, such as "DARTS[1]", "P-DARTS[2]" and "PC-DARTS[3]".

## Appendix J. Experimental details of evaluating robustness against overfitting

The four search spaces $S1 - S4$ are designed by (Zela et al., 2020).

---

1. https://github.com/quark0/darts

2. https://github.com/chenxin061/pdarts

3. https://github.com/yuhuixu1993/PC-DARTS/

- **S1**: In this search space, each edge has only two candidate operations. To identify these operations, operations that have the least importance in the original search space of DARTS are iteratively removed.

- **S2**: For each edge, the candidate operations are 3×3 SepConv and SkipConnect.

- **S3**: For each edge, the candidate operations are: 3×3 SepConv, SkipConnect, and Zero.

- **S4**: For each edge, the candidate operations are: 3×3 SepConv and Noise. In the Noise operation, every value from the input feature map is replaced with random variables sampled from univariate Gaussian distribution.

## Appendix K. Significance test results

To check whether the performance of our proposed methods are significantly better than baselines, we perform a statistical significance test using a double-sided T-test. We use the function in the python package "scipy.stats.ttest_1samp" and report the average results over 10 different runs. Table 5 and 6 show the results.

| Our method | Baseline | p-value |
|---|---|---|
| Ours-darts2nd | GMPGC-darts2nd | 3.65e-3 |
| Ours-darts2nd | CDEP-darts2nd | 6.29e-5 |
| Ours-darts2nd | EC-darts2nd | 1.40e-6 |
| Ours-darts2nd | Darts2nd | 5.51e-10 |
| Ours-pcdarts | GMPGC-pcdarts | 2.64e-5 |
| Ours-pcdarts | CDEP-pcdarts | 3.59e-6 |
| Ours-pcdarts | EC-pcdarts | 1.35e-5 |
| Ours-pcdarts | Pcdarts | 2.59e-7 |
| Ours-prdarts | GMPGC-prdarts | 6.33e-3 |
| Ours-prdarts | CDEP-prdarts | 1.27e-3 |
| Ours-prdarts | EC-prdarts | 5.49e-4 |
| Ours-prdarts | Prdarts | 3.41e-3 |
| Ours-pdarts | GMPGC-pdarts | 5.75e-8 |
| Ours-pdarts | CDEP-pdarts | 9.83e-9 |
| Ours-pdarts | EC-pdarts | 6.62e-7 |
| Ours-pdarts | Pdarts | 8.57e-9 |

Table 5: Significance test results on CIFAR-100

From these two tables, we can see that the p-values are small between baselines methods and our methods, which demonstrate that the errors of our methods are significantly lower than those of baselines.

## Appendix L. Additional visualization of saliency maps

In this section, we present additional visualization of the saliency maps generated for some images (Figure 3) in ImageNet.

| Our method | Baseline | p-value |
|---|---|---|
| Ours-darts2nd | GMPGC-darts2nd | 4.46e-7 |
| Ours-darts2nd | CDEP-darts2nd | 6.82e-5 |
| Ours-darts2nd | EC-darts2nd | 7.49e-8 |
| Ours-darts2nd | Darts2nd | 1.25e-5 |
| Ours-pcdarts | GMPGC-pcdarts | 6.64e-6 |
| Ours-pcdarts | CDEP-pcdarts | 2.71e-7 |
| Ours-pcdarts | EC-pcdarts | 9.39e-6 |
| Ours-pcdarts | Pcdarts | 5.05e-5 |
| Ours-prdarts | GMPGC-prdarts | 6.47e-5 |
| Ours-prdarts | CDEP-prdarts | 7.53e-6 |
| Ours-prdarts | EC-prdarts | 9.72e-6 |
| Ours-prdarts | Prdarts | 2.88e-4 |
| Ours-pdarts | GMPGC-pdarts | 8.51e-6 |
| Ours-pdarts | CDEP-pdarts | 2.49e-7 |
| Ours-pdarts | EC-pdarts | 4.73e-8 |
| Ours-pdarts | Pdarts | 9.58e-5 |

Table 6: Significance test results on CIFAR-10

For each image, our method performs a PGD adversarial attack to generate a perturbation matrix $\delta$. The size of $\delta$ is the same as image size, where $\delta_{ij}$ denotes the perturbation at pixel $(i, j)$. Then we take the element-wise absolute value of $\delta$ and get $|\delta|$. Pixels with larger values in $|\delta|$ are more important. We use a heatmap to visualize $|\delta|$ where pixels with warmer colors (e.g., red) are more important and are more suitable to be used as explanations.

Figure 4 shows the saliency maps and Figure 5 shows the overlay of saliency maps on top of original images. As can be seen, the highly salient regions (warmer color regions) in our method are very sensible. They correspond to objects. In contrast, the colder color (e.g., blue) regions correspond to background. These results show that our method is effective in generating correct saliency maps.

## Appendix M. Details of hyperparameters

### M.1. Adversarial attack settings

To calculate the perturbations $\delta$, we use 7 iterations PGD (Madry et al., 2017) attack with settings shown in Table 7.

### M.2. Full lists of hyperparameter settings

Table 9, Table 10, Table 11, and Table 12 show the hyperparameter settings of our methods during architecture search on CIFAR-10 and CIFAR-100. Table 13 shows the hyperparameter settings of Ours-PCDARTS during architecture search on ImageNet. Table 14 and Table 15 show the hyperparameter settings in the architecture evaluation phrase. Notations used in these tables are given in Table 8.

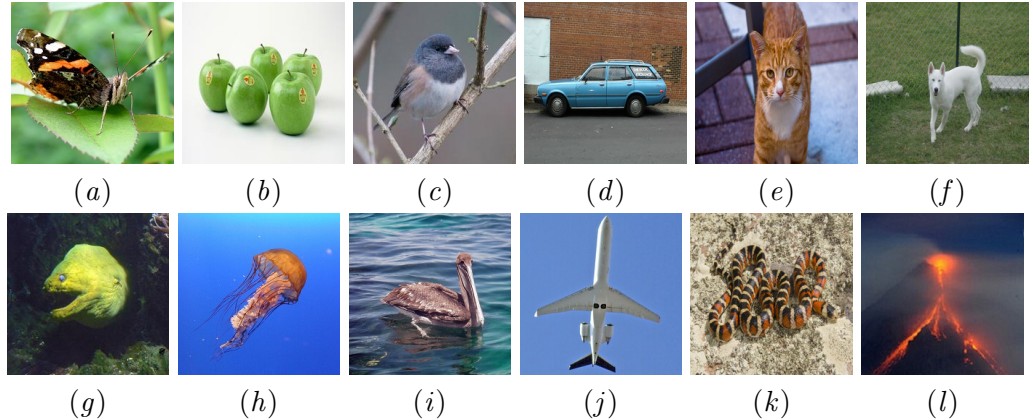

Figure 3: Original images.

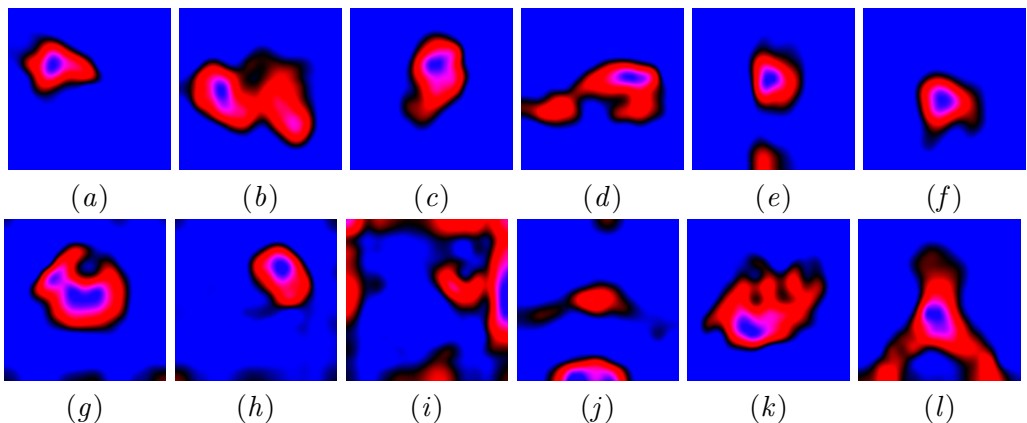

Figure 4: Heatmaps of perturbations (absolute value).

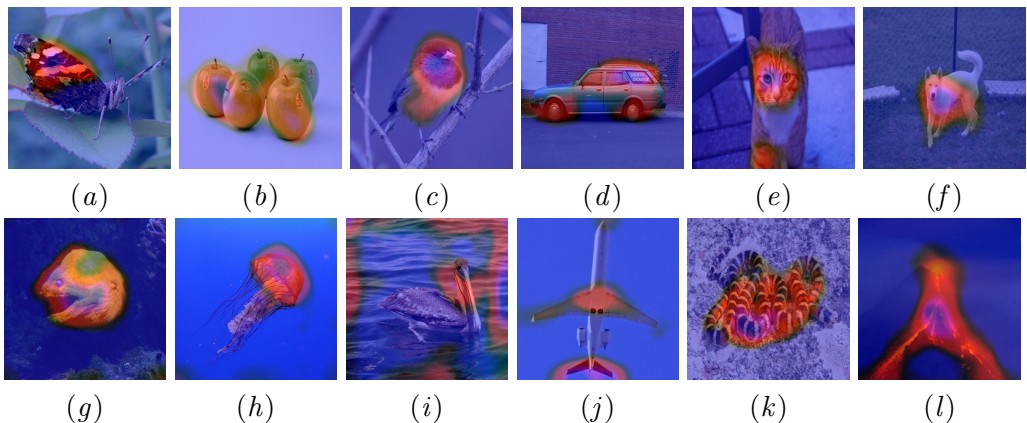

Figure 5: Overlay of saliency maps to original images.

| PGD | |
|---|---|
| Iterations | 7 |
| Perturbation $\epsilon$ | 0.03 (8/255) |
| Step size | 2/255 |
| Norm | $l_\infty$ |

Table 7: Hyperparameters for PGD attack to obtain perturbations $\delta$.

Table 8: Notations in WSEDP

| Notation | Meaning |
|---|---|
| $A$ | Architecture |
| $W_1$ | Network weights of the first model |
| $W_2$ | Network weights of the second model |
| $D^{(\mathrm{tr})}$ | Training data |
| $D^{(\mathrm{val})}$ | Validation data |

| Name | Value |
|---|---|
| **Hyperparameters for training network weights $W_1$ and $W_2$** | |
| Initial learning rate | 0.025 |
| Minimum learning rate | 0 |
| Epochs | 50 |
| Momentum | 0.9 |
| Weight decay | 3e-4 |
| Batch size | 64 |
| Gradient clipping | 5 |
| Initial channels | 16 |
| Layers | 8 |
| **Hyperparameters for learning architecture variables $A$** | |
| Optimizer | Adam |
| Learning rate | $3e-4$ |
| Weight decay | $1e-3$ |
| $\beta$ | (0.5, 0.999) |

Table 9: Hyperparameters for Ours-DARTS2nd during architecture search on CIFAR-10 and CIFAR-100.

| Name | Value |
|---|---|
| **Hyperparameters for training network weights $W_1$ and $W_2$** | |
| Initial learning rate | 0.025 |
| Minimum learning rate | 0 |
| Epochs | 50 |
| Momentum | 0.9 |
| Weight decay | 3e-4 |
| Batch size | 64 |
| Gradient clipping | 5 |
| Initial channels | 16 |
| Layers | 6 |
| **Hyperparameters for learning architecture variables $A$** | |
| Optimizer | Adam |
| Learning rate | $3e-4$ |
| Weight decay | $1e-3$ |
| $\beta$ | (0.5, 0.999) |

Table 10: Hyperparameters for Ours-PCDARTS during architecture search on CIFAR-10 and CIFAR-100.

| Name | Value |
|---|---|
| **Hyperparameters for training network weights $W_1$ and $W_2$** | |
| Initial learning rate | 0.025 |
| Minimum learning rate | 0 |
| Epochs | 50 |
| Momentum | 0.9 |
| Weight decay | 3e-4 |
| Batch size | 64 |
| Gradient clipping | 5 |
| Initial channels | 16 |
| Layers | 5,11,17 |
| **Hyperparameters for learning architecture variables $A$** | |
| Optimizer | Adam |
| Learning rate | $3e-4$ |
| Weight decay | $1e-3$ |
| $\beta$ | (0.5, 0.999) |

Table 11: Hyperparameters for Ours-PDARTS during architecture search on CIFAR-10 and CIFAR-100.

| Name | Value |
|---|---|
| **Hyperparameters for training network weights $W_1$ and $W_2$** | |
| Initial learning rate | 0.025 |
| Minimum learning rate | 0 |
| Epochs | 50 |
| Momentum | 0.9 |
| Weight decay | 3e-4 |
| Batch size | 64 |
| Gradient clipping | 5 |
| Initial channels | 16 |
| Layers | 5,11,17 |
| **Hyperparameters for learning architecture variables $A$** | |
| Optimizer | Adam |
| Learning rate | $3e-4$ |
| Weight decay | $1e-3$ |
| $\beta$ | (0.5, 0.999) |

Table 12: Hyperparameters for Ours-PRDARTS during architecture search on CIFAR-10 and CIFAR-100.

| Name | Value |
|---|---|
| **Hyperparameters for training network weights $W_1$ and $W_2$** | |
| Learning rate | 0.5 |
| Minimum learning rate | 0 |
| Epochs | 50 |
| Momentum | 0.9 |
| Weight decay | 3e-4 |
| Batch size | 1024 |
| Gradient clipping | 5 |
| Initial channels | 16 |
| Layers | 8 |
| **Hyperparameters for learning architecture variables $A$** | |
| Optimizer | Adam |
| Learning rate | $3e-4$ |
| Weight decay | $1e-3$ |
| $\beta$ | (0.5, 0.999) |

Table 13: Hyperparameters for Ours-PCDARTS during architecture search on ImageNet.

| Name | Value |
|---|---|
| Optimizer | SGD |
| Learning rate | 0.025 |
| Minimum learning rate | 0 |
| Epochs | 600 |
| Momentum | 0.9 |
| Weight decay | 3e-4 |
| Initial channels | 36 |
| Batch size | 96 |
| Number of layers (cells) | 20 |
| Gradient clipping | 5 |
| Auxiliary weight | 0.4 |
| Cutout length | 16 |
| Initial drop probability | 0.3 |

Table 14: Hyperparameters during architecture evaluation on CIFAR-10 and CIFAR-100.

| Name | Value |
|---|---|
| Optimizer | SGD |
| Learning rate | 0.5 |
| Minimum learning rate | 0 |
| Epochs | 250 |
| Momentum | 0.9 |
| Weight decay | 3e-5 |
| Initial channels | 48 |
| Batch size | 1024 |
| Number of layers (cells) | 14 |
| Gradient clipping | 5 |
| Auxiliary weight | 0.4 |
| Initial drop probability | 0 |

Table 15: Hyperparameters during architecture evaluation on ImageNet.

## Appendix N. Architectures visualization

Figures 6 to 13 show the normal cells and reduction cells searched using our methods on CIFAR-10 and CIFAR-100.

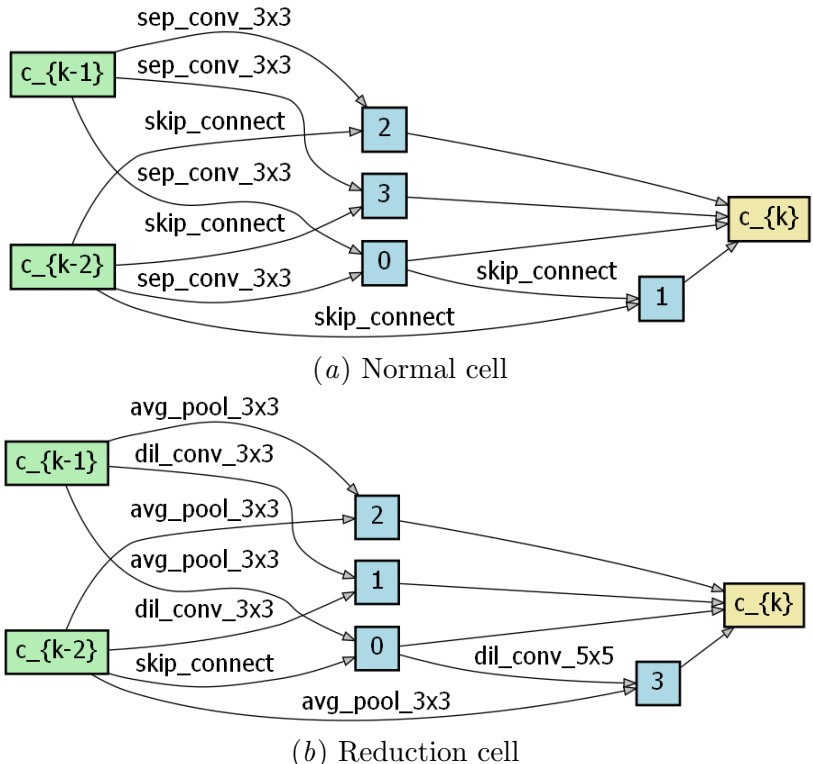

(*a*) Normal cell

(*b*) Reduction cell

Figure 6:  Cells searched by Ours-DARTS2nd on CIFAR-10.

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

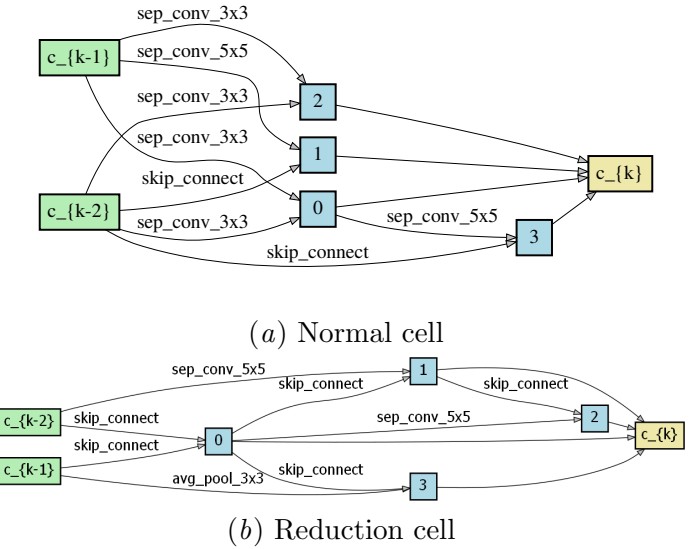

(*a*) Normal cell

(*b*) Reduction cell

Figure 7: Cells searched by Ours-PDARTS on CIFAR-10.

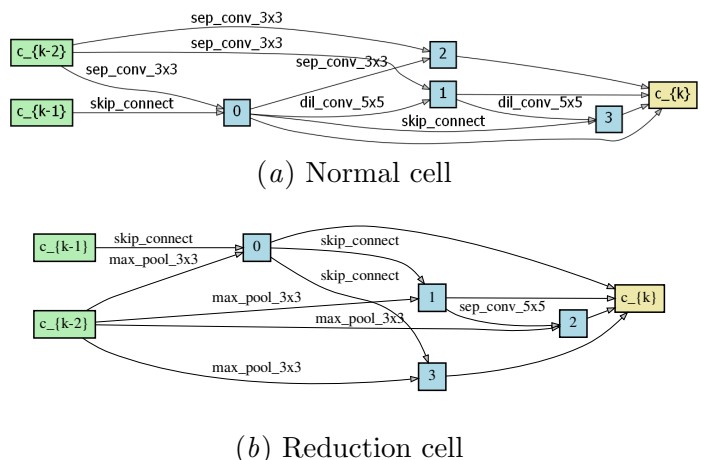

(*a*) Normal cell

(*b*) Reduction cell

Figure 8: Cells searched by Ours-PCDARTS on CIFAR-10.

Xiangxiang Chu, Tianbao Zhou, Bo Zhang, and Jixiang Li. Fair DARTS: eliminating unfair advantages in differentiable architecture search. *CoRR*, abs/1911.12126, 2019.

Xiangxiang Chu, Xiaoxing Wang, Bo Zhang, Shun Lu, Xiaolin Wei, and Junchi Yan. DARTS-: robustly stepping out of performance collapse without indicators. *CoRR*, abs/2009.01027, 2020a.

Xiangxiang Chu, Bo Zhang, and Xudong Li. Noisy differentiable architecture search. *CoRR*, abs/2005.03566, 2020b.

Xuanyi Dong and Yi Yang. Searching for a robust neural architecture in four GPU hours. In *CVPR*, 2019.

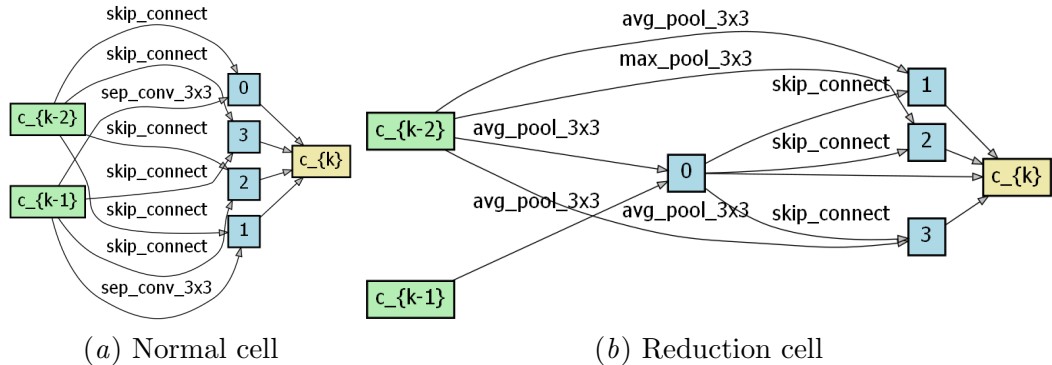

(a) Normal cell          (b) Reduction cell

Figure 9: Cells searched by Ours-PRDARTS on CIFAR-10.

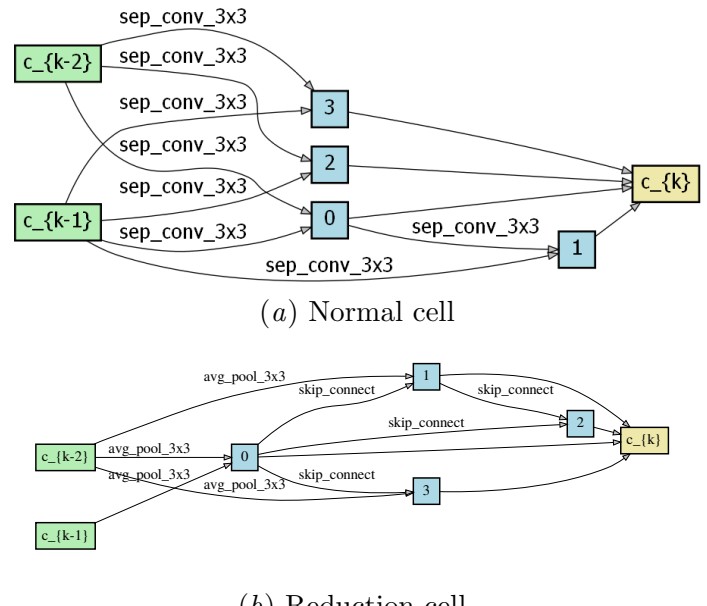

(a) Normal cell

(b) Reduction cell

Figure 10: Cells searched by Ours-DARTS2nd on CIFAR-100.

Saeed Ghadimi and Mengdi Wang. Approximation methods for bilevel programming. *arXiv preprint arXiv:1802.0f2246*, 2018.

Riccardo Grazzi, Luca Franceschi, Massimiliano Pontil, and Saverio Salzo. On the iteration complexity of hypergradient computation. In *International Conference on Machine Learning*, pages 3748–3758. PMLR, 2020.

Andrew G. Howard, Menglong Zhu, Bo Chen, Dmitry Kalenichenko, Weijun Wang, Tobias Weyand, Marco Andreetto, and Hartwig Adam. Mobilenets: Efficient convolutional neural networks for mobile vision applications. *CoRR*, abs/1704.04861, 2017.

Shoukang Hu, Sirui Xie, Hehui Zheng, Chunxiao Liu, Jianping Shi, Xunying Liu, and Dahua Lin. DSNAS: direct neural architecture search without parameter retraining. In *CVPR*,

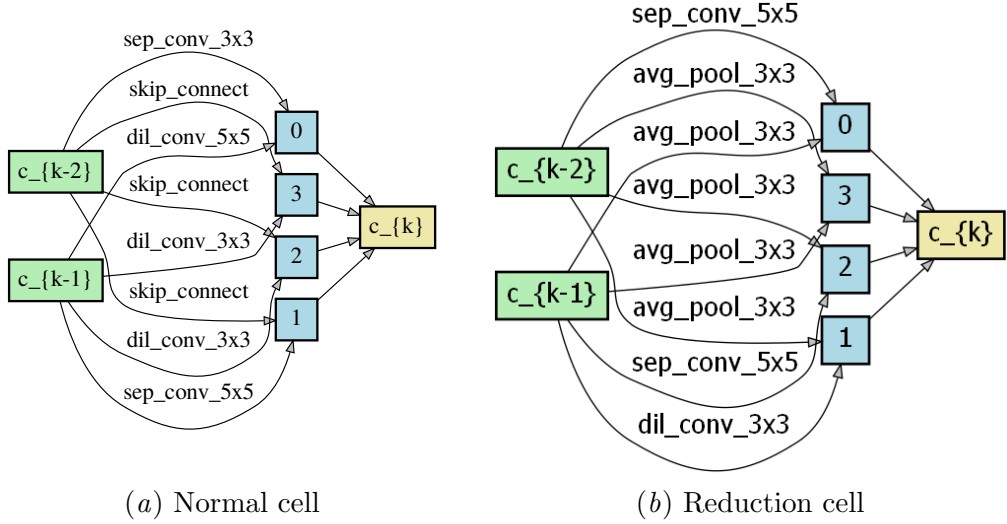

$(a)$ Normal cell $\qquad\qquad$ $(b)$ Reduction cell

Figure 11: Cells searched by Ours-PDARTS on CIFAR-100.

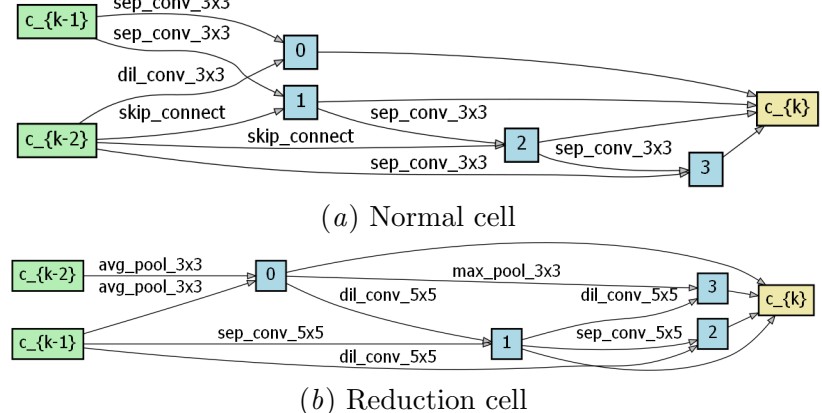

$(a)$ Normal cell

$(b)$ Reduction cell

Figure 12: Cells searched by Ours-PCDARTS on CIFAR-100.

2020.

Kaiyi Ji, Junjie Yang, and Yingbin Liang. Bilevel optimization: Convergence analysis and enhanced design. In *International Conference on Machine Learning*, pages 4882–4892. PMLR, 2021.

Diederik P Kingma and Jimmy Ba. Adam: A method for stochastic optimization. *arXiv preprint arXiv:1412.6980*, 2014.

Hanwen Liang, Shifeng Zhang, Jiacheng Sun, Xingqiu He, Weiran Huang, Kechen Zhuang, and Zhenguo Li. DARTS+: improved differentiable architecture search with early stopping. *CoRR*, abs/1909.06035, 2019.

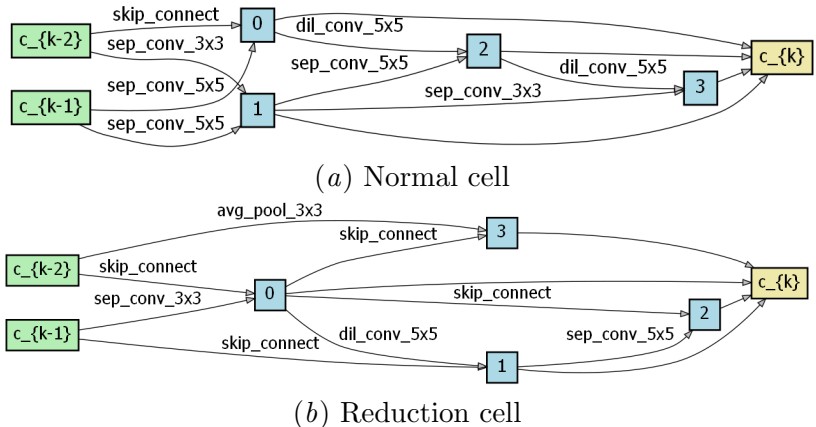

(a) Normal cell

(b) Reduction cell

Figure 13: Cells searched by Ours-PRDARTS on CIFAR-100.

Chenxi Liu, Barret Zoph, Maxim Neumann, Jonathon Shlens, Wei Hua, Li-Jia Li, Li Fei-Fei, Alan L. Yuille, Jonathan Huang, and Kevin Murphy. Progressive neural architecture search. In *ECCV*, 2018.

Hanxiao Liu, Karen Simonyan, and Yiming Yang. DARTS: differentiable architecture search. In *ICLR*, 2019.

Risheng Liu, Yaohua Liu, Shangzhi Zeng, and Jin Zhang. Towards gradient-based bilevel optimization with non-convex followers and beyond. *Advances in Neural Information Processing Systems*, 34, 2021.

Ningning Ma, Xiangyu Zhang, Hai-Tao Zheng, and Jian Sun. Shufflenet V2: practical guidelines for efficient CNN architecture design. In *ECCV*, 2018.

Aleksander Madry, Aleksandar Makelov, Ludwig Schmidt, Dimitris Tsipras, and Adrian Vladu. Towards deep learning models resistant to adversarial attacks. *arXiv preprint arXiv:1706.06083*, 2017.

Paulius Micikevicius, Sharan Narang, Jonah Alben, Gregory Diamos, Erich Elsen, David Garcia, Boris Ginsburg, Michael Houston, Oleksii Kuchaiev, Ganesh Venkatesh, et al. Mixed precision training. *arXiv preprint arXiv:1710.03740*, 2017.

Esteban Real, Alok Aggarwal, Yanping Huang, and Quoc V Le. Regularized evolution for image classifier architecture search. In *Proceedings of the aaai conference on artificial intelligence*, volume 33, pages 4780–4789, 2019.

Christian Szegedy, Wei Liu, Yangqing Jia, Pierre Sermanet, Scott Reed, Dragomir Anguelov, Dumitru Erhan, Vincent Vanhoucke, and Andrew Rabinovich. Going deeper with convolutions. In *CVPR*, 2015.

Mingxing Tan, Bo Chen, Ruoming Pang, Vijay Vasudevan, Mark Sandler, Andrew Howard, and Quoc V. Le. Mnasnet: Platform-aware neural architecture search for mobile. In *CVPR*, 2019.

Ronald J Williams. Simple statistical gradient-following algorithms for connectionist reinforcement learning. *Machine learning*, 8(3):229–256, 1992.

Sirui Xie, Hehui Zheng, Chunxiao Liu, and Liang Lin. SNAS: stochastic neural architecture search. In *ICLR*, 2019.

Yuhui Xu, Lingxi Xie, Xiaopeng Zhang, Xin Chen, Guo-Jun Qi, Qi Tian, and Hongkai Xiong. PC-DARTS: partial channel connections for memory-efficient architecture search. In *ICLR*, 2020.

Junjie Yang, Kaiyi Ji, and Yingbin Liang. Provably faster algorithms for bilevel optimization. *Advances in Neural Information Processing Systems*, 34, 2021.

Arber Zela, Thomas Elsken, Tonmoy Saikia, Yassine Marrakchi, Thomas Brox, and Frank Hutter. Understanding and robustifying differentiable architecture search. In *ICLR*, 2020.

Xiangyu Zhang, Xinyu Zhou, Mengxiao Lin, and Jian Sun. Shufflenet: An extremely efficient convolutional neural network for mobile devices. In *CVPR*, 2018.

Hongpeng Zhou, Minghao Yang, Jun Wang, and Wei Pan. Bayesnas: A bayesian approach for neural architecture search. In *ICML*, 2019.

Pan Zhou, Caiming Xiong, Richard Socher, and Steven Hoi. Theory-inspired path-regularized differential network architecture search. In *Neural Information Processing Systems*, 2020.

Barret Zoph and Quoc V. Le. Neural architecture search with reinforcement learning. In *ICLR*, 2017.

Barret Zoph, Vijay Vasudevan, Jonathon Shlens, and Quoc V Le. Learning transferable architectures for scalable image recognition. In *CVPR*, 2018.