# OpenReview forum: "Saliency-Aware Neural Architecture Search"
_NeurIPS.cc/2022/Conference — NeurIPS 2022 Accept_

### Official Review · Reviewer_sfyZ · 2022-06-28

**Rating:** 6
**Confidence:** 3
**Soundness:** 2 fair
**Presentation:** 3 good
**Contribution:** 2 fair

**Summary:**

The paper unifies a four step optimization solution for obtaining a neural architecture, considering the salience of data.

**Questions:**

There is a relevant line of research in pruning of neural networks using explanations that maps to similar salience maps. It would be worth looking into that area as there seems a nice bridge between the two line of research.

**Limitations:**

No specific limitation discussed

**Strengths And Weaknesses:**

The paper is well written and provides a well articulated explanation of the solution and results. The paper provides sufficient amount of experiments that support the usefulness of the proposed approach. I do not find a particular weakness for this work.

---

> ### Author Response · Authors · 2022-08-02
> **Author response to Reviewer sfyZ**
>
> We would like to thank the reviewer for the positive and constructive feedback.
>
> 1. Thanks for mentioning the line of research on saliency-based network pruning. We looked at some papers in this field, which is indeed an interesting direction that our work can be bridged with, in  future studies. Specifically, we plan to investigate the following ideas: 1) formulate saliency-based network pruning as a saliency-aware neural architecture search problem and automatically search for the optimal pruning decisions based on detected saliency; 2) extend the notion of saliency from input data to blocks in neural networks, develop multi-level optimization based framework to detect the saliency of blocks, and perform pruning on blocks based on detected saliency.
>
>    We added such discussions in Lines 403-410 in the rebuttal revision.
>
>
> 2. For the limitations of our method, we discussed them in the last section in the initial submission and further expanded the discussion in Lines 387-399 and 656-683 in the rebuttal revision.

---

> > ### Comment · Reviewer_sfyZ · 2022-08-10
> > **response to author feedback**
> >
> > Thank you for your feedback. The paper seems a good one to be further discussed officially in the community. I am raising the score.

---

> > > ### Author Response · Authors · 2022-08-10
> > > **Thank you for reading our response and raising the score.**
> > >
> > > We thank the reviewer for reading our response and raising the score. We highly appreciate the reviewer's constructive suggestions.

---

### Official Review · Reviewer_vRUp · 2022-07-11

**Rating:** 6
**Confidence:** 3
**Soundness:** 3 good
**Presentation:** 3 good
**Contribution:** 3 good

**Summary:**

This paper leverages the classic differentiable architecture search strategy for neural architecture search through a complex optimization process that involves training that reweights architectural parameters and weights according to a strategy where pixel values are biased by a saliency score that derives from the degree of effect on accuracy subject to perturbation of pixel values.

**Questions:**

1. For the four level optimization framework, this appears to be very complex but the cost values in the results tables suggests it is very efficient. Could the authors comment a bit more on what makes this approach so efficient given an apparently highly complex loss function?

2. In table 7, how does one judge which words should be deemed most salient? Anecdotally what is shown seems to make sense, but further clarity on how to judge this would be interesting.

3. The paper admits that for evolutionary or reinforcement based methods that the approach can't be used. However, is it possible to extend the basic notion of saliency that is presented here to be adapted to such a method? If not, what is the central limitation?

**Limitations:**

The authors have discussed both limitations and potential for negative societal impact which is minimal.

**Strengths And Weaknesses:**

Strengths:

1. The work successfully demonstrates that the proposed saliency measure in the context of the optimization carried out in [40] gives rise to improved neural architectures.

2. The optimization procedure, albeit complex, is formulated in a way that is relatively straightforward to optimize outside of the role that hyperparameter sensitivity may play in the process.

Weaknesses:

1. It's not clear to what degree the hyperparameter tuning is easy/difficulty to achieve although the ablation studies and extra analysis presented shows some confidence for general purpose use of this method.

2. Visualization of saliency maps in figure 2 could be more visible/clear.

3. The conclusion focuses too much on the limitations/weaknesses and should restate the central contributions.

---

> ### Author Response · Authors · 2022-08-02
> **Author response to Reviewer vRUp**
>
>
>
> We would like to thank the reviewer for the positive and constructive feedback. In the submitted rebuttal revision paper, we have addressed the weaknesses mentioned by the reviewer. The updates are marked with blue color. We summarize how these weaknesses are addressed and answer the reviewer's questions below.
>
> 1. **Q**: Comment on what makes the proposed approach efficient.
>
>     **A**: We improved computational efficiency from both the algorithm side and implementation side.
>
>      On the algorithm side, we speed up computation by approximating the optimal solution at each stage using a one-step gradient descent update and reducing the update frequencies of some parameters. Specifically, we update the architecture $A$ every 5 mini-batches, update model weights $W_2$ and perturbations $\delta$ every 3 mini-batches, and update $W_1$ on every mini-batch. We empirically found that reducing the update frequencies of $A$, $W_2$, and $\delta$ greatly speeds up convergence without significantly sacrificing performance.
>      Besides, when calculating hypergradients of $A$, we recursively approximate matrix-vector multiplications using finite-difference calculations, which reduces the computation cost from being quadratic in matrix dimensions down to linear.
>
>     On the implementation side, we speed up computation by leveraging techniques and tricks including 1) automatic mixed precision, 2) using multiple (4, specifically) workers and pinned memory in PyTorch DataLoader, 3) using cudNN autotuner, 4) kernel fusion, etc.
>
>     We added these discussions in Lines 694-707 in the rebuttal revision.
>
> $~$
>
>
> 2. **Q**: In table 7, how does one judge which words should be deemed most salient?
>
>    **A**: The prediction task corresponding to this table is sentiment classification. A word is more salient if it has a stronger correlation with a sentiment. For example, the word “entertaining” implies a positive sentiment, and therefore is considered to be salient. We added these discussions in Lines 689-693 in the rebuttal revision.
>
> $~$
>
>
>
> 3. **Q**: Is it possible to extend the basic notion of saliency to be adapted to evolutionary or reinforcement learning based methods?
>
>     **A**: It is possible to perform saliency-aware architecture search in reinforcement learning (RL) based NAS methods, as follows. First, use an RL controller to generate a set of candidate architectures. Second, given a candidate architecture, train its weight parameters on a training dataset. Third, given the trained model, perform adversarial attacks to detect the saliency maps of the training data, similar to stage II in our method. Fourth, use saliency maps to reweight training data and retrain the model on reweighted data. Fifth, evaluate the retrained model on a validation set and use validation accuracy as a reward of this architecture. Repeat steps 2-5 for every candidate architecture, calculate the mean reward on all candidate architectures, and update the RL controller by maximizing the mean reward using policy gradient. These procedures repeat until convergence. Similar procedures can be conducted to perform saliency-aware architecture search in evolutionary algorithm based NAS methods. In these approaches, gradient-based optimization cannot be used any more since the objectives are not differentiable.
>
>     We added these discussions in Lines 657-671 in the rebuttal revision.
>
>
> $~$
>
>
>
> 4. **Q**: To what degree the hyperparameter tuning is easy/difficult to achieve?
>
>      **A**: Most hyperparameters in our method follow their default values used in baseline methods. The only    hyperparameter needing to be tuned is the tradeoff parameter $\gamma$. To tune $\gamma$ on CIFAR-100, we randomly sample 2.5K data from the 25K training set and sample 2.5K data from the 25K validation set. Then we use the 5K sampled data as a hyperparameter tuning set. $\gamma$ is tuned in {0.1, 0.5, 1, 2, 3}. For each configuration of $\gamma$, we use the remaining 22.5K training data and 22.5K validation data to perform architecture search and use their combination to perform architecture evaluation (retraining a larger stacked network from scratch). Then we measure the performance of the stacked network on the 5K sampled data. $\gamma$ value yielding the best performance is selected. For $\gamma$ in CIFAR-10 and ImageNet experiments, we simply used the value tuned on CIFAR-100 and didn’t conduct further tuning.
>
>      We added these discussions in Lines 708-718 in the rebuttal revision.
>
> $~$
>
>
>
> 5. **Q**: Make the visualization of saliency maps in Figure 2 be more visible/clear.
>
>      **A**: In rebuttal revision, we enlarged the size of saliency maps and added original images in Figure 3 and 5 to make the visualization of saliency maps more visible.
>
> $~$
>
>
>
> 6.  **Q**: Restate the central contributions in the conclusion section.
>
>      **A**: In rebuttal revision, we have added more discussions of the central contributions in Lines 376-386.

---

### Official Review · Reviewer_Am7u · 2022-07-12

**Rating:** 7
**Confidence:** 4
**Soundness:** 3 good
**Presentation:** 4 excellent
**Contribution:** 3 good

**Summary:**

Saliency-Aware Neural Architecture Search presents a four-step neural architecture search (NAS) procedure that reweights inputs based on their saliency value:
1. The original architecture is trained on the original dataset.
2. Using the trained model, an adversarial saliency method generates saliency maps for all inputs. The features of the inputs are reweighted based on their saliency value.
3. The original architecture is retrained on the saliency weighted inputs.
4. The model trained on the saliency weighted inputs is evaluated on the validation dataset. A new loss-minimizing architecture is selected using a traditional differentiable NAS method (e.g., DARTS).

Each of the four steps is repeated until convergence.

The saliency-aware neural architecture search is evaluated on image classification tasks using CIFAR-10, CIFAR-100, and ImageNet and on text classification tasks using GLUE. The proposed method is combined with other differentiable NAS methods like DARTS-2nd, PC-DARTS, PR-DARTS, and P-DARTS. It is evaluated against non-NAS models, NAS-selected models, and other saliency-based training procedures to understand changes in performance, training time, and robustness. The saliency results are visualized and evaluated based on their alignment with human expectations.

**Questions:**

1. What would happen if you switched the adversarial saliency method in step 3 for other types of saliency methods like integrated gradients, SmoothGrad, GradCAM, LIME, Sufficient Input Subsets, etc.? Would SANAS apply to nondifferentiable NAS methods if you used a non-gradient-based saliency method like LIME or Sufficient Input Subsets?
2. How does the adversarial saliency method do on existing saliency map tests (see Sanity Checks for Saliency Maps by Adebayo et al.)?

**Limitations:**

Can you please expand on other limitations of your method? Should I always use SANAS instead of a different differentiable NAS approach?

For instance, there seems to be a time cost of SANAS (e.g., Table 3 lists 1.1 GPU days compared to P-DARTS 0.3 GPU days). How should someone trade off between time cost and potential performance benefit?

Section 5 mentions that SANAS can not easily be applied to nondifferentiable NAS methods. Why is this and what are the implications? Are there ways to use it with nondifferentiable NAS methods (e.g., if you used a non-gradient based saliency method like LIME)?

**Strengths And Weaknesses:**

**Strengths**
* *Evaluative methods* ---  The paper does a comprehensive evaluation of the SANAS method. They evaluate text and image modalities, various NAS methods, NAS and non-NAS models, and related saliency-based training algorithms. In addition, they include robustness tests and an ablation study on saliency reweighting.
* *Evaluative results* --- The results show that SANAS results in models with lower error rates while maintaining reasonable parameter and time costs. SANAS also results in better robustness to overfitting and prevents performance collapse.
* *Clarity and Reproducibility* --- The paper and appendix are very well written and thorough. The method was easy to follow. All necessary details are included, including hyperparameter settings, model parameters, optimization algorithms, human study details, implementation details, and code packages for significant testing. It would be easy to reproduce.

**Weaknesses**
* *Limitations and societal implications* --- The limitations and societal implications section of this paper lacks detail. An expanded limitations section would help readers understand when to use SANAS over an alternative method and the specific benefits of SANAS.
* *Saliency evaluation* --- The paper evaluates its saliency maps by visualizing them to human users who judge their alignment with human expectations. The paper states that visualizing the saliency maps shows that the "method is effective in generating correct saliency maps." However, this evaluation assumes the model has learned the same features humans have (i.e., the saliency should highlight the object of interest). While ideally, the model would learn features that align with human expectations, research has shown that models often learn to rely on spurious features. Saliency methods should be faithful to the model's representations. If the model's representations do not align with a human's, then the saliency should not be sensible to a human interpreter. Other research has shown that sensible-looking saliency methods can mask the underlying model's behavior and inhibit proper interpretation of the results. The main contribution of the paper is not a new saliency method. Still, if the authors evaluate the saliency maps, they should use saliency map tests such as model parameter randomization tests or data randomization tests (see Sanity Checks for Saliency Maps by Adebayo et al.).
* *Qualitative saliency results* --- The text examples show an example text saliency map. One example is not sufficient to claim the SANAS method is better than EC and GMPGC. Please include additional examples in the appendix.
* *Saliency methods* --- The paper only uses an adversarial saliency method in its framework. The experiments evaluate against other saliency-based training procedures (EC, CDEP, and GMPGC) that use different saliency methods. However, these procedures differ from SANAS because they add regularization terms to the loss function instead of reweighting the dataset. Comparing to other saliency methods would strengthen the paper. If you found similarly positive results, it would validate that the benefit of SANAS generalizes beyond a single saliency method. However, even if these methods do not work as well, it would be interesting scientifically and shed light on potential differences between adversarial and non-adversarial methods.

**Minor Comments**
* Table 1: please explain what the asterisks represent.
* Line 138: "bottom to up" --> "bottom-up"
* Line 71 and 253: please expand on the unreliability of GradCAM or cite prior work.

---

> ### Author Response · Authors · 2022-08-02
> **Author response to Reviewer Am7u**
>
> We'd like to thank the reviewer for the positive and constructive feedback. In the submitted rebuttal revision paper, we have addressed the weaknesses mentioned by the reviewer. We summarize how these weaknesses are addressed and answer the reviewer's questions below.
>
> 1. **Q**: Experiment with other saliency methods.
>
>    **A**: In the rebuttal revision (Table 6 and Lines 306-318), we experimented with two more saliency methods: integrated gradients and SmoothGrad. The table below shows the results. Our framework with IntegratedGrad and SmoothGrad still outperforms  Darts2nd and Pdarts. This demonstrates that our framework is a general one that generalizes beyond a single saliency method. Second, IntegratedGrad and SmoothGrad perform worse than Adversarial Saliency. A possible reason is: IntegratedGrad and SmoothGrad restrict the definition of saliency to be gradient-based. In contrast, Adversarial treats saliency scores as optimization variables and automatically learns them by solving an optimization problem, which is more flexible.
>
>      | Method        | Error on CIFAR-100 | Error on CIFAR-10|
>      | ------------- |-------------| -----|
>      |Darts2nd| 20.58±0.44 | 2.76±0.09 |
>      |IntegratedGrad-Darts2nd| 16.92±0.08 | 2.62±0.06 |
>      |SmoothGrad-Darts2nd| 17.05±0.11 | 2.59±0.03 |
>      |Adversarial-Darts2nd| **16.42**±0.09| **2.54**±0.05|
>      |||||
>      |Pdarts| 17.52±0.06 | 2.54±0.04 |
>      |IntegratedGrad-Pdarts| 15.83±0.08 | 2.47±0.03 |
>      |SmoothGrad-Pdarts| 15.81±0.05 | 2.48±0.04 |
>      |Adversarial-Pdarts| **15.16**±0.09| **2.45**±0.03|
>      |||||
>
> $~$
>
>
> 2. **Q**: Would SANAS apply to nondifferentiable NAS methods? Why SANAS can not easily be applied to nondifferentiable NAS methods and what are the implications?
>
>    **A**: By changing the gradient-based optimization algorithm to non-gradient-based algorithm, we can apply SANAS to nondifferentiable NAS methods, such as reinforcement learning (RL) based NAS methods, as follows. First, use an RL controller to generate candidate architectures. Second, given a candidate architecture, train it on a training dataset. Third, given the trained model, perform adversarial attacks to detect saliency maps of training data. Fourth, use saliency maps to reweight training data and retrain the model on reweighted data. Fifth, evaluate the retrained model on a validation set and use validation accuracy as a reward of this architecture. Repeat steps 2-5 for every candidate architecture, calculate the mean reward, and update the RL controller by maximizing the mean reward. These procedures repeat until convergence. Similar procedures can be conducted to perform saliency-aware NAS in evolutionary algorithm based NAS methods.
>
>       The **reason that SANAS can not easily be applied to nondifferentiable NAS methods** is that  SANAS uses a gradient-based optimization algorithm to solve the multi-level optimization problem. For nondifferentiable NAS methods, their nondifferentiable objective functions do not have gradients.
>
>      The **implication** is: if we want to apply SANAS to nondifferentiable NAS methods, we have to change the gradient-based optimization algorithm to some other non-gradient-based algorithms.
>
>      We added these discussions in Lines 388-393 and 657-671 in rebuttal revision.
>
> $~$
>
>
> 3. **Q**: Perform saliency map tests for the adversarial saliency method.
>
>    **A**: In the rebuttal revision (Lines 249-258), we evaluated saliency maps using model parameter cascading randomization tests. Figure 2(left) in the rebuttal revision shows that saliency maps change considerably  as more layers are randomized. Figure 2(right) shows the Spearman rank correlation between original saliency maps and  randomized saliency maps consistently decreases as more layers are randomized. These  results demonstrate that saliency maps generated by the adversarial saliency method are sensitive to model parameters and pass the sanity check.
>
> $~$
>
>
>
> 4. **Q**: Discuss other limitations of SANAS. When to use SANAS and when not? How to trade off between time cost and performance?
>
>    **A**: In the rebuttal revision (Lines 388-399 and 656-683), we expanded the limitation section. As pointed out by the reviewer, one limitation of our method is the time cost. Another downside is that SANAS is mathematically more complicated than baselines, which adds some extra difficulty for usage.
>
>     It is recommended to use SANAS in applications that strongly need high-performance architectures capable of generating sensible saliency maps but do not have strong efficiency requirements on architecture search time. For applications which have high restrictions on search cost but allow sacrificing some performance and ignoring saliency maps,  other NAS methods might be better choices.
>
>
> $~$
>
>
> 5.  We added four more examples of text saliency in Table 10 (on page 17) in rebuttal revision.
>
> 6.  We addressed comments marked as minor in rebuttal revision.

---

> > ### Comment · Reviewer_Am7u · 2022-08-07
> > **Rebuttal Revision LGTM**
> >
> > I want to thank the authors for incorporating the suggestions from the other reviewers and me. They have done significant work adding new experiments and evaluations that strengthen the paper. I am satisfied with all of the changes, and I continue to think this paper should be accepted.

---

> > > ### Author Response · Authors · 2022-08-07
> > > **Thank you**
> > >
> > > We would like to thank the reviewer for reading our response and supporting the acceptance of our paper. The constructive and valuable suggestions from the reviewer and other reviewers help to improve this paper a lot. We highly appreciate these suggestions, which are incorporated into the rebuttal revision.

---

### Official Review · Reviewer_X4zW · 2022-07-16

**Rating:** 5
**Confidence:** 4
**Soundness:** 3 good
**Presentation:** 3 good
**Contribution:** 3 good

**Summary:**

This paper proposes an end-to-end framework which dynamically detects saliency of input data, reweights data using saliency maps, and searches architectures on saliency-reweighted data. The proposed framework is based on four-level optimization, which performs four learning stages in a unified way. Experiments on several datasets demonstrate the effectiveness of the proposed framework.

**Questions:**

Please see the above commons.

**Limitations:**

Please see the above commons.

**Strengths And Weaknesses:**

This paper is interesting. It tries to address the limitation in existing NAS methods which treat all data elements as being equally important. Experiments show that the proposed NAS method achieves good performance.

At the end of the introduction part, the third contributions should be removed, which is a common part of a scientific paper rather than a special contribution.

There should be some ablation studies to validate the effectiveness of various designs (like the four stages) in the proposed method. This is important for better understanding the proposed method.

---

> ### Author Response · Authors · 2022-08-02
> **Author response to Reviewer X4zW**
>
> We would like to thank the reviewer for the positive and constructive feedback. In the submitted rebuttal revision paper, we have addressed the weaknesses mentioned by the reviewer. The updates are marked with blue color. We summarize how these weaknesses are addressed and answer the reviewer's questions below.
>
> 1. **Q**: The third contribution at the end of the introduction section should be removed.
>
>      **A**: In the rebuttal revision, we have removed the third contribution from the introduction section.
>
>  $~$
>
> 2. **Q**: There should be some ablation studies to validate the effectiveness of various designs.
>
>     **A**: In our initial submission, we reported ablation studies on 1) saliency reweighting mechanisms in Lines 285-301 of the main paper, 2) sensitivity on the hyperparameter $\gamma$ in Lines 302-306 of the main paper, and 3) performing stage I and II by minimizing a single objective in Section 5 of the supplementary material.
>
>     **In the rebuttal revision (Table 6 and 8, and Lines 306-334), we added three more ablation studies**, which 1) perform the four stages separately (denoted as Separate) instead of end-to-end; 2) perform stages 1-3 by minimizing the weighted sum of their objective functions in a multi-task learning (MTL) way (denoted as MTL); and 3) compare the adversarial attack based saliency detection method with other saliency detection methods including Integrated Gradient (IntegratedGrad) and SmoothGrad.  These studies were performed on Darts2nd and Pdarts.
>    **The table below shows  results on Separate and MTL**. From this table, we make two observations. First, our end-to-end method works better than Separate which conducts the four stages separately. Conducting the four stages end-to-end can enable them to mutually influence each other to achieve the best overall performance. In contrast, when conducted separately, earlier stages cannot be influenced by later stages (e.g., stage I cannot be influenced by stage IV), which leads to worse performance. Second, our method performs better than MTL. The tasks in stages I-III have an inherent order: before detecting saliency maps using a model, we first need to train this model; before training the second model on saliency-reweighted data, we need to detect the saliency maps first. MTL performs these three tasks simultaneously by minimizing a single objective, which breaks their inherent order and therefore leads to worse performance. In contrast, our method preserves this order using multi-level optimization.
>
>     | Method        | Error on CIFAR-100 | Error on CIFAR-10|
>     | ------------- |-------------| -----|
>     |Separate-Darts2nd| 18.05±0.27 | 2.68±0.06 |
>     |MTL-Darts2nd| 18.26±0.12 |2.70±0.05 |
>     |Ours-Darts2nd| **16.42**±0.09| **2.54**±0.05|
>     |||||
>     |Separate-Pdarts| 16.49±0.07 |2.51±0.03 |
>     |MTL-Pdarts| 16.83±0.10 |2.52±0.04 |
>     |Ours-Pdarts| **15.16**±0.09| **2.45**±0.03|
>     |||||
>
>      **The table below shows results on IntegratedGrad and SmoothGrad**, where we make two observations. First,  our framework with IntegratedGrad and SmoothGrad as saliency detection methods still outperforms  Darts2nd and Pdarts. This demonstrates that our framework is a general one that generalizes beyond a single saliency detection method. Second, IntegratedGrad and SmoothGrad perform worse than Adversarial. A possible reason is: IntegratedGrad and SmoothGrad restrict the definition of saliency to be gradient-based. In contrast, Adversarial treats saliency scores as optimization variables and automatically learns them by solving an optimization problem, which is more flexible.
>
>      | Method        | Error on CIFAR-100 | Error on CIFAR-10|
>      | ------------- |-------------| -----|
>      |Darts2nd| 20.58±0.44 | 2.76±0.09 |
>      |IntegratedGrad-Darts2nd| 16.92±0.08 | 2.62±0.06 |
>      |SmoothGrad-Darts2nd| 17.05±0.11 | 2.59±0.03 |
>      |Adversarial-Darts2nd| **16.42**±0.09| **2.54**±0.05|
>      |||||
>      |Pdarts| 17.52±0.06 | 2.54±0.04 |
>      |IntegratedGrad-Pdarts| 15.83±0.08 | 2.47±0.03 |
>      |SmoothGrad-Pdarts| 15.81±0.05 | 2.48±0.04 |
>      |Adversarial-Pdarts| **15.16**±0.09| **2.45**±0.03|
>      |||||

---

### Meta-Review · Area_Chair_dcLY · 2022-08-26

**Recommendation:** Accept
**Confidence:** Certain

**Metareview:**

This paper proposed a novel method that reweights data using saliency maps and searches architecture using saliency-reweighted data.

There are four official reviewers for this submission. The reviewers consistently agree with the novelty, presentation, and experimental validation of this submission. The ratings are: borderline accept/accept/weak accept. The concerns raised by the reviewers are well addressed during the rebuttal.

Thus the AC would like to recommend acceptance.



**Award:**

No

---

### Decision · Program_Chairs · 2022-09-14

Accept